

# From climatological to small scale applications: Simulating water isotopologues with ICON-ART-Iso (version 2.1)

Johannes Eckstein[1], Roland Ruhnke[1], Stephan Pfahl[2], Emanuel Christner[1], Christoph Dyroff[1,a], Daniel Reinert[3], Daniel Rieger[3], Matthias Schneider[1], Jennifer Schröter[1], Andreas Zahn[1], and Peter Braesicke[1]

[1]Karlsruhe Institute of Technology (KIT), Institute of Meteorology and Climate Research (IMK), Herrmann-von-Helmholtz-Platz 1, 76344 Eggenstein-Leopoldshafen, Germany
[2]Institute for Atmospheric and Climate Science, ETH Zurich, 8092 Zurich, Switzerland
[3]Deutscher Wetterdienst, Frankfurter Str. 135, 63067 Offenbach, Germany
[a]now at: Aerodyne Research Inc., 45 Manning Road, Billerica, MA 01821, USA

*Correspondence to:* Johannes Eckstein (johannes.eckstein@kit.edu)

**Abstract.** We present the new isotope enabled model ICON-ART-Iso. The physics of the global ICOsahedral Nonhydrostatic (ICON) modelling framework have been extended to simulate passive moisture tracers and the stable isotopologues HDO and $H_2^{18}O$. The extension builds on the infrastructure provided by ICON-ART, which allows a high flexibility with respect to the number of related water tracers that are simulated. The physics of isotopologue fractionation follow the model COSMOiso.

First, we present a detailed description of the physics of fractionation that have been implemented in the model. The model is then evaluated by comparing with measurements in precipitation and vapor representing a range of temporal scales.

A multi annual simulation is compared to observations of the isotopologues in precipitation taken from the station network GNIP (Global Network for Isotopes in Precipitation). ICON-ART-Iso is able to reasonably simulate the seasonal cycles in $\delta D$ and $\delta^{18}O$ as observed at the GNIP stations. In a comparison with IASI satellite retrievals, the seasonal and daily cycles in

the isotopologue content of vapor are examined for different regions in the free troposphere. On a small spatial and temporal scale, ICON-ART-Iso is used to simulate the period of two flights of the IAGOS-CARIBIC aircraft in September 2010, which sampled air in the tropopause level influenced by Hurricane Igor. The general features of this sample as well as all of tropical data available from IAGOS-CARIBIC are captured by the model.

The study demonstrates that ICON-ART-Iso is a flexible tool to analyze the water cycle of ICON. It is capable of simulating

tagged water as well as the isotopologues HDO and $H_2^{18}O$.

## 1 Introduction

Water in gas, liquid and frozen form is an important component of the climate system. The ice caps and snow covered surfaces strongly influence the albedo of the surface (Kraus, 2004), the oceans are unmatched reservoir to dissolve trace substances (Jacob, 1999) and redistribute heat (Pinet, 1993) and all animal and plant life depends on liquid water. The atmosphere is by

mass the smallest compartment of the hydrological cycle, but it is this compartment that serves to transfer water between the spheres of liquid, frozen and biologically bound water on the earth's surface (Gat, 1996). For atmospheric processes themselves,





water is also of great importance. It is the strongest green house gas (Schmidt et al., 2010), distributes energy through the release of latent heat (Holton and Hakim, 2013) and liquid and frozen particles influence the radiative balance (Shine and Sinha, 1991), to name only three prominent mechanisms.

A correct description of the atmospheric water cycle is therefore necessary for the understanding and simulation of the
atmosphere and the climate system (Riese et al., 2012; Sherwood et al., 2014). The stable isotopologues of water are unique diagnostic tracers that provide a deeper insight into the water cycle (Galewsky et al., 2016). Because of the larger molar mass of the heavy isotopologues, their ratio to (standard) water is changed by phase transitions, a process termed fractionation. By considering the heavy isotopologues in vapor and precipitation (liquid or ice), the isotopologue ratio therefore provides an opportunity to develop an advanced understanding of the processes that shape the water cycle.

Pioneering research on measuring the heavy isotopologues of water starting in the 1950's first examined the isotopologues in precipitation (Dansgaard, 1954, 1964). First theoretical advances on the microphysics (Jouzel et al., 1975; Jouzel and Merlivat, 1984) and surface evaporation (Craig and Gordon, 1965) enabled the implementation of heavy isotopologues in global climate models (Joussaume et al., 1984; Joussaume and Jouzel, 1993). Since then, measurement techniques and modeling of the isotopologues have advanced. Cryogenic samplers first measured the isotopologue content in vapor (Dansgaard, 1954), which
has become possible by laser absorption spectroscopy in the last 15 years (Lee et al., 2005; Dyroff et al., 2010). Today, the isotopologue content in atmospheric vapor can also be derived from satellite measurements (Gunson et al., 1996; Worden et al., 2006; Steinwagner et al., 2007; Schneider and Hase, 2011). Many global and regional circulation models have been equipped to simulate the atmospheric isotopologue distribution, either focusing on climatological questions (Risi et al., 2010; Werner et al., 2011) or process understanding (Blossey et al., 2010; Pfahl et al., 2012, both limited area models). Despite this progress,
the potential of isotopologues in improving the understanding and physical description of the single processes "remains largely unexplored" (Galewsky et al., 2016). A more extensive literature overview on the subject is given by Galewsky et al. (2016).

We present ICON-ART-Iso, the newly developed, isotopologue enabled version of the global ICOsahedral Nonhydrostatic (ICON) modelling framework (Zängl et al., 2015). By design, ICON is a flexible model, capable of simulations from climatological down to turbulent scales (Heinze et al., 2017). Klocke et al. (2017) show the potential of using ICON for convection
permitting simulations. The advection scheme of ICON has been designed to be mass conserving (Zängl et al., 2015), which is essential for the simulation of water isotopologues (Risi et al., 2010; Cauquoin and Risi, 2017). ICON-ART-Iso builds on the flexible infrastructure provided by the extension ICON-ART (Rieger et al., 2015; Schröter et al., 2017, in preparation), which has been developed to simulate aerosols and trace gases.

By equipping ICON with the capabilities to simulate the water isotopologues, a first step is made to a deeper understanding
of the water cycle. From the multitude of isotopologue enabled models (see Galewsky et al. (2016) for an overview), ICON-ART-Iso stands out by its non-hydrostatic base model core, enabling simulations with fine horizontal resolution on a global grid. It is flexible in design to simulate diagnostic moisture tracers (also termed tagged water) as well as the isotopologues HDO and $H_2^{18}O$ during a single simulation.

This article first gives some technical details on ICON and ICON-ART. This is followed by a detailed description of the
physics special to ICON-ART-Iso, which have been implemented in ICON to simulate the isotopologues (Sec. 2).





The remaining sections describe model results and first validation studies: Section 3.1 looks at precipitation diagnostics. Focus is laid on the source regions - ocean or land - of the precipitating water. The next section (Sec. 3.2) compares data from a simulation spanning more than seven years on a coarse grid to measurements from different stations of the GNIP network. A further validation with measurements is performed in Sec. 3.3. Retrievals from IASI satellite measurements are compared with

ICON-ART-Iso results for two weeks in winter and summer 2014, considering the seasonal and daily cycle in different regions. Section 3.4 then discusses the comparison with IAGOS-CARIBIC measurements. In situ data from two flights are compared with simulations of ICON-ART-Iso. Section 4 summarizes and concludes the study.

## 2 The model ICON-ART-Iso

This section presents the technical and physical background of the model ICON-ART-Iso. First, ICON and the extension

ICON-ART are introduced. Next, general thoughts on simulating a diagnostic water cycle are presented. Starting in Sec. 2.3, the main processes that influence the distribution of the isotopologues are discussed in separate sections: surface evaporation, saturation adjustment, cloud microphysics and convection. Finally, Sec. 2.7 discusses the initialization of the model.

### 2.1 Introduction to the modeling framework ICON-ART

ICON-ART-Iso is the isotope enabled version of the model ICON. ICON is a new non-hydrostatic general circulation model

which is developed and maintained in a joint effort by Deutscher Wetterdienst (DWD) and Max-Planck-Institute for Meteorology (MPI-M). Its horizontally unstructured grid can be refined locally by one-way or two-way nested domains with a higher resolution. The model is applicable from global to turbulent scales. At DWD, ICON is used operationally for global numerical weather prediction, it already proved successful as a Large Eddy Simulation (LES) model (Heinze et al., 2017), and is currently prepared for climate prediction studies at MPI-M. More details on ICON are given by Zängl et al. (2015).

ICON-ART-Iso builds on the numerical weather prediction physics parameterization package of ICON. The physical parameterizations that have been implemented for the simulation of the isotopologues correspond to those of the model COSMOiso as presented by Pfahl et al. (2012). As the same parameterizations have been described before, the following subsections give only a short summary of each of the different fractionation processes.

In ICON, all tracer constituents are given as mass fractions $q_x = \frac{\rho_x}{\rho}$, where $\rho = \sum_x \rho_x$ is the total density, including all water

constituents $x$. To discriminate values the heavy isotopologues, these will be denoted by the index $h$ while standard (light) water will be indexed by $l$. ICON standard water is identified with the light isotopologue $H_2O$, which is a very good assumption also made by Pfahl et al. (2012) and Blossey et al. (2010). Water in ICON-ART-Iso exists in seven different forms, each of which is represented by one tracer for standard water and each of the isotopologues: vapor, cloud water, ice, rain, snow, graupel and hail. The amount of the isotopologues is expressed relative to standard water by the isotopologue ratio $R = {}^h q_x / {}^l q_x$. This is

referenced to standard ratios of the Vienna Mean Ocean Standard Water ($R_{\text{VSMOW}}$) in the $\delta$ notation: $\delta = R_{\text{sample}} / R_{\text{VSMOW}} - 1$, with $\delta$ values then given in permil.





As in the current version of ICON-ART (Schröter et al., 2017, in preparation), an XML table is used to define the settings for each of the isotopologues. This makes the model very flexible and allows using several different water tracers during one model run, where each of them makes use of different fractionation parameterizations.

## 2.2 Simulating a diagnostic water cycle

In general, the isotopologues are affected by all those processes that also influence standard water in ICON: Surface evaporation, saturation adjustment to form clouds, cloud microphysics and convection. Each of these main processes is described by several parameterizations. Some of these parameterizations include phase changes of or to vapor, which lead to a change in the isotopologue ratio - which is termed isotopic fractionation. In addition, advection and turbulent diffusion are non-fractionating processes that change the spatial distribution of all trace substances.

An important prerequisite to a simulation of water isotopologues is a good implementation of advection (Cauquoin and Risi, 2017). ICON-ART makes use of the same numerical methods that are used for advecting the hydrometeors in ICON itself. These assure local mass conservation (Zängl et al., 2015) and mass-consistent transport. The latter is achieved by making use of the same mass flux in the discretized continuity equations for total density and partial densities, respectively (Lauritzen et al., 2014). The advection schemes implemented in ICON conserve linear correlations between tracers and assure the monotonicity

of each advected tracer. Note, however, that this does not gaurantee monotonicity of the isotopologue ratios (see Morrison et al., 2016).

The parameterizations influence the water cycle also include processes that do not fractionate. For all non-fractionating processes, the transfer rate $^hS$ of the heavier isotopologues are defined by Eq. 1.

$$^hS = {^lS} \cdot \frac{^hq_x^{\text{source}}}{^lq_x^{\text{source}}} \tag{1}$$

Here, $q_x$ denotes the mass ratio of the hydrometeor $x$, which is taken from the source of the process. $^lS$ is the transfer rate of ICON standard water in $q_x$. This equation can also be applied to processes that would normally change the isotopologue ratio in order to turn off fractionation. This has been implemented as an option in all processes that describe fractionation. If all processes are set to be non-fractionating, the isotopologue ratio does not change and the species will resemble the standard water in ICON. This is an important feature which can be used to test the model for self-consistency or to investigate source

regions with so called tagged water (e.g. Bosilovich and Schubert, 2002). An application of this will be shown in Sec. 3.1.

Whenever phase changes including the vapor phase occur, the isotopologue ratio changes because of the different diffusion constants and the different saturation vapor pressure of the heavier isotopologues. For the diffusion constant ratio, two choices have been implemented for HDO and $H_2{}^{18}O$, using the values of Merlivat and Jouzel (1979) or Cappa et al. (2003). The differences in saturation pressure are expressed with the equilibrium fractionation factor $\alpha$, which is the ratio of isotopologues

ratios in thermodynamic equilibrium. This ratio $\alpha$ depends on temperature and is different over water and over ice (termed $\alpha_{\text{liq}}$ and $\alpha_{\text{ice}}$). For all parameterizations, Eq. 2 holds, where $R_v$ stands for the isotopologue ratio in the vapor phase, while $R_{\text{cond}}$ stands for that in the condensed phase. The parameterizations by Majoube (1971) and by Horita and Wesolowski (1994) have been implemented for $\alpha_{\text{liq}}$ and those by Merlivat and Nief (1967) for $\alpha_{\text{ice}}$. The definition for $\alpha$ given in Eq. 2 is also used in



COSMOiso (Pfahl et al., 2012), but is the inverse of the definition by Blossey et al. (2010).

$$\alpha = \frac{R_v}{R_{\mathrm{cond}}} < 1 \tag{2}$$

## 2.3 Surface evaporation

Surface evaporation is the source for the atmospheric water cycle. In ICON-ART-Iso, the evaporative surface flux is split into evaporation from land and water surfaces, transpiration from plants and dew and rime formation. Transpiration is considered a non-fractionating process (Eq. 1), which is an assumption also made by Werner et al. (2011) or Pfahl et al. (2012). Dew and rime formation (and condensation on the ocean surface) are considered to fractionate according to equilibrium fractionation (Eq. 2). For the evaporation part of the full surface flux, two parameterizations have been implemented (Pfahl and Wernli, 2009; Merlivat and Jouzel, 1979). Both build on the Craig-Gordon model (Craig and Gordon, 1965; Gat, 2010). Equation 3 gives the general expression for $R_{\mathrm{evap}}$.

$$R_{\mathrm{evap}} = k \cdot \frac{\alpha_{\mathrm{liq}} R_{\mathrm{surf}} - h R_v}{1 - h} \tag{3}$$

Here, $h$ is the specific humidity of the lowest model layer relative to the specific humidity at the surface and $k$ is the non-equilibrium fractionation factor. The two parameterization differ in their description of $k$. While Merlivat and Jouzel (1979) give a parameterization that depends on the surface wind, Pfahl and Wernli (2009) have simplified this to be wind speed independent. In summary, Eq. 4 is used to calculate the surface flux of the isotopologues, $^h F^{tot}$.

$$^h F^{tot} = {}^l F^{\mathrm{evap}} \cdot R_{\mathrm{evap}} + {}^l F^{\mathrm{transp}} \cdot R_{\mathrm{surf}} + {}^l F^{\mathrm{dew}} \cdot \frac{R_v}{\alpha_{\mathrm{liq}}} + {}^l F^{\mathrm{rime}} \cdot \frac{R_v}{\alpha_{\mathrm{ice}}} \tag{4}$$

For transpiration and evaporation, the isotopologue ratio of the surface and ground water ($R_{\mathrm{surf}}$) is necessary. The surface model TERRA of ICON was not equipped with isotopologues for ICON-ART-Iso, so $R_{\mathrm{surf}}$ is not available as a prognostic variable. It is therefore approximated by $R_{\mathrm{VSMOW}}$ in Eq. 3 and 4. While this may be regarded as a valid approximation over the ocean (there are only small variations, see LeGrande and Schmidt, 2006), it can only be regarded as a first order step over land. Nevertheless, it is common procedure in most isotopologue enabled atmospheric models (e.g. Risi et al., 2010; Werner et al., 2011).

## 2.4 Saturation adjustment

Cloud water is formed by saturation adjustment in ICON. Vapor in excess of saturation vapor pressure is transferred to cloud water and temperature is adjusted accordingly. This is repeated in an iterative procedure. For the isotopologues, the iteration does not have to be repeated. Instead, Eq. 5 is applied directly, using the adjusted values of ICON water. This is the same equation used in COSMOiso (Pfahl et al., 2012) and by Blossey et al. (2010).

$$^h q_c = \frac{^h q_v + {}^h q_c}{1 + \alpha_{\mathrm{liq}} \frac{^l q_v}{^l q_c}} \tag{5}$$



## 2.5 Microphysics

Several grid-scale microphysical schemes are available in ICON. ICON-ART-Iso makes use of the two moment scheme by Seifert and Beheng (2006). It computes mass and number densities of vapor, cloud water, rain and four ice classes (ice, snow, graupel and hail) and can be used to simulate aerosol-cloud interaction, see Rieger et al. (2017). As the isotopologues are

diagnostic values, the number densities do not have to be simulated separately. The two moment scheme describes more than 60 different processes, but only those processes that include the vapor phase ilead to fractionation. All others are described by Eq. 1. In accordance with Blossey et al. (2010) and Pfahl et al. (2012), sublimation is also assumed not to fractionate. Condensation to form liquid water happens only during the formation of cloud water and is accounted for by the saturation adjustment. The fractionating processes that remain are ice formation by nucleation (of ice), deposition (on all four ice classes)

and evaporation of liquid hydrometeors. Besides rain, a fraction of the three larger ice classes (snow, graupel, hail) can evaporate after melting. This liquid water fraction is currently not a prognostic variable.

The two moment scheme by Seifert and Beheng (2006) uses mass densities instead of mass ratios, so we adopt the change in notation here, denoting mass densities by $\rho$. The star ($^*$) indicates values at saturation with respect to liquid (index $l$) or ice (index $i$).

For evaporation, two parameterizations have been implemented to describe fractionation. One choice is the semi-empirical parameterization by Stewart (1975) also implemented in COSMOiso (Pfahl et al., 2012). The change in mass, $^hS_x^{\text{evap}}$, is calculated from $^lS_x^{\text{evap}}$ as given in Eq. 6.

$$^hS_x^{\text{evap}} = {}^lS_x^{\text{evap}} \left(\frac{^hD}{^lD}\right)^n \frac{\alpha_{\text{liq}} {}^l\rho_{l,v}^* \frac{^h\rho_x}{^l\rho_x} - {}^h\rho_v}{^l\rho_{l,v}^* - {}^l\rho_v} \tag{6}$$

Here, $\rho_x$ stands for the mass density of either rain or melting snow, graupel or hail while $\rho_v$ indicates the mass density of the

vapor phase. The ratio of the diffusion constants $D$ is given by the literature values cited above. The tuning parameter $n$ is set to 0.58 by default (see again Stewart, 1975).

Another option to describe fractionation during evaporation of hydrometeors is the parameterization following the theoretical approach by Blossey et al. (2010), given in Eq. 7.

$$^hS_x^{\text{evap}} = 4\pi a\, {}^hf \frac{\left((1+b_l)\frac{R_v}{R_{\text{hyd}}}\alpha_{\text{liq}} - b_l\right) {}^lS_l - 1}{\frac{\alpha_{\text{liq}}}{\zeta R_{\text{hyd}}}\left(\frac{\mathcal{R}_v T_\infty}{^lD e_{l,\infty}^*} + \frac{L_e^2}{k_a \mathcal{R}_v T_\infty^2}\right)} \tag{7}$$

$$b_l = \frac{^lD\, L_e^2\, \rho_{l,\infty}^*}{k_a \mathcal{R}_v T_\infty^2} \tag{8}$$

The equation is derived from fundamentals of cloud microphysics (see Pruppacher and Klett, 2012), with the adaptation for the isotopologues explained in detail by Blossey et al. (2010). The same equations are used to derive $^lS_x^{\text{evap}}$ in the microphysical scheme in ICON (Seifert and Beheng, 2006). Here, $a$ is the radius of the hydrometeor, $^hf$ is the ventilation factor (set equal to $^lf$ in default setup), $R_v$ and $R_{\text{hyd}}$ are the isotopologue ratios in the vapor and the hydrometeor, $\mathcal{S}_l$ the subsaturation ratio,

$\zeta = {}^hD/^lD$ the ratio of diffusion constants, $\mathcal{R}_v$ the gas constant of water vapor, $T_\infty$ and $e_{l,\infty}^*$ the temperature and saturation vapor pressure in the surroundings and $L_e$ and $k_a$ are the latent heat of evaporation and the heat conductivity.



Fractionation during nucleation of ice particles or deposition on one of the four ice classes is parameterized following Blossey et al. (2010), as in COSMOiso (Pfahl et al., 2012). The flux is assumed to interact only with the outermost layer of the hydrometeor, the isotopologue ratio of which is set to be identical to that of the depositional flux. The transfer rate $^hS_x^{\text{ice}}$ is then given by Eq. 9 with the fractionation factor $\alpha_k$ as given in Eq. 10. All symbols are used as in Eq. 7, with $L_s$ being the latent heat of sublimation.

$$^hS_x^{\text{ice}} = \alpha_k R_v \, ^lS_x^{\text{ice}} \tag{9}$$

$$\alpha_k = \frac{(1+b_i)\, ^lS_i}{^hf \frac{1}{\zeta}\left(^lS_i - 1\right) + \alpha_{\text{ice}}\left(1 + b_i \, ^lS_i\right)} \tag{10}$$

$$b_i = \frac{^lD_v \, L_s \, \rho_{i,\infty}^*}{k_a \, \mathcal{R}_v \, T_\infty^2} \tag{11}$$

### 2.6 Convection

ICON uses the Tiedtke-Bechtold scheme for simulating convective processes (Tiedtke, 1989; Bechtold et al., 2014). The scheme uses a simple cloud model considering a liquid fraction in cloud water (denoted here by $\omega$) and the remaining solid fraction $(1-\omega)$. Fractionation happens during convective saturation adjustment (during initialization of convection and in updrafts), in saturated downdrafts and in evaporation below cloud base. The parameterizations are the same that have been implemented by Pfahl et al. (2012) in COSMOiso.

Convective saturation adjustment calculates equilibration between vapor and the total condensed water (liquid and ice). The parameterization used for grid scale adjustment therefore has to be expanded in order to be used in convection if the liquid water fraction is smaller than one. The isotopologue ratio is determined over liquid and ice particles separately. A closed system approach (Gat, 1996) is used for the liquid fraction ($R_v^{\text{by liq}}$ of Eq. 12). The underlying assumption for Eq. 13 used for the ice fraction is a Rayleigh process with the kinetic fractionation factor $\alpha_{\text{eff}}$ following Jouzel and Merlivat (1984). The two are then recombined according to the fraction of liquid water, following Eq. 14. This procedure has been adopted from COSMOiso (Pfahl et al., 2012).

$$R_v^{\text{by liq}} = R_v^{\text{old}} \frac{\alpha_{\text{liq}}}{1 + \frac{^lq_v^{\text{new}}}{^lq_v^{\text{old}}}\left(\alpha_{\text{liq}} - 1\right)} \tag{12}$$

$$R_v^{\text{by ice}} = R_v^{\text{old}} \left(\frac{^lq_v^{\text{new}}}{^lq_v^{\text{old}}}\right)^{\alpha_{\text{eff}} - 1} \tag{13}$$

$$R_v = (1-\omega) \cdot R_v^{\text{by ice}} + \omega \cdot R_v^{\text{by liq}} \tag{14}$$

Here, the indices `old` and `new` denote the values of the respective variables before and after the convective saturation adjustment. The factor $\alpha_{\text{eff}}$ which appears in Eq. 13 is determined by Eq. 15. The supersaturation with respect to ice, $\xi_{\text{ice}}$, is calculated from Eq. 16, where $T_0 = 273.15\,\text{K}$ is used. The tuning parameter $\lambda$ is set to 0.004 in the standard setup, following Pfahl et al.

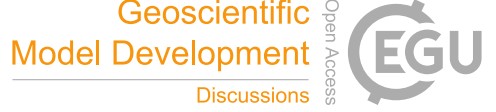

(2012) and Risi et al. (2010).

$$\alpha_{\text{eff}} = \frac{\xi_{\text{ice}}\zeta}{\xi_{\text{ice}} - 1 + \alpha_{\text{ice}}\zeta} \tag{15}$$

$$\xi_{\text{ice}} = 1 - \lambda(T - T_0) \tag{16}$$

Convective downdrafts are assumed to remain saturated by continuously evaporating precipitation (Tiedtke, 1989). Equilib-
rium fractionation is applied for the liquid fraction, while the ice fraction is assumed to sublimate without fractionation.

Evaporation of precipitation below cloud base leads to a drop in the temperature and is therefore an important process
in convection. To describe fractionation here, the parameterization by Stewart (1975) is again applied to the liquid fraction.
Different to Eq. 6 for evaporation during microphysics, the integrated form is now applied. In following Stewart (1975), the
ratio in the liquid part of the general hydrometeor after evaporation $R_{\text{adj}}^{\text{liq}}$ is given with Eq. 17. Here, $f$ is the fraction of
remaining condensate. $R_{\text{hyd}}^{\text{old}}$ is the isotopologue ratio in the hydrometeor before adjustment and rH is the relative humidity
calculated as the vapor pressure over saturation vapor pressure.

$$R_{\text{adj}}^{\text{liq}} = \gamma R_v + f^\beta \left( R_{\text{hyd}}^{\text{old}} - \gamma R_v \right) \tag{17}$$

$$\gamma = \frac{\text{rH}}{\alpha_{\text{liq}} - \mu} \tag{18}$$

$$\beta = \frac{\alpha_{\text{liq}} - \mu}{\mu} \tag{19}$$

$$\mu = (1 - \text{rH}) \left( \frac{{}^hD}{{}^lD} \right)^{-n} \tag{20}$$

Using Eq. 17, the isotopologue ratio in the adjusted hydrometeor is given with Eq. 21. The ice fraction is assumed to sublimate
without fractionation, maintaining its isotopologue ratio.

$$R_{\text{adj}} = (1 - \omega)R_{\text{hyd}}^{\text{old}} + \omega R_{\text{adj}}^{\text{liq}} \tag{21}$$

Following Pfahl et al. (2012), an additional equilibration has been implemented to determine the final isotopologue ratio of the
hydrometeors, which is given in Eq. 22. The parameter $\xi_{\text{add}}$ is a tuning parameter that is set to 0.5 in standard setup.

$$R_{\text{adj}}^{\text{final}} = R_{\text{adj}} + \xi_{\text{add}} \cdot \omega \left( \frac{R_v}{\alpha_{\text{liq}}} - R_{\text{adj}} \right) \tag{22}$$

### 2.7 General initialization of the isotopologues

The first step necessary for simulating the isotopologues is a meaningful initialization. In addition to an initialization with a
constant ratio to standard water, which is interesting for simulating tagged water, the isotopologues can be initialized with the
help of mean measured $\delta$ values. Values at the lowest model level, the tropopause level (WMO definition, see Holton et al.,
1995) and model top are prescribed for vapor and linear and log-linear interpolation is applied below and above the tropopause,
respectively. Values for the tropopause level and the model top are taken from MIPAS measurements (Steinwagner et al., 2007)
and given in Table 1. By using the local tropopause height, an adaptation to the local meteorological situation is assured. To
calculate the $\delta$ value of the hydrometeors, a constant offset is applied to the local $\delta$ value of vapor. Values for $H_2{}^{18}O$ are
determined from the relationship given by the global meteoritic water line (GMWL), $\delta D = 8 \cdot \delta^{18}O + 10‰$ (Craig, 1961).




|  | literature | HDO | $H_2^{18}O$ |
|---|---|---:|---:|
| $\delta_{\text{bottom}}$ | Gat (2010) | -50 | -5 |
| $\delta_{\text{tropopause}}$ | Steinwagner et al. (2007) | -650 | -80 |
| $\delta_{\text{top}}$ | Steinwagner et al. (2007) | -400 | -48.75 |
| $\delta_{\text{offset}}$ | Gat (2010) | -100 | -11.25 |

**Table 1.** Values for the initialization with mean measured $\delta$ values. Literature sources give the values for HDO, values for $H_2^{18}O$ have been determined from GMWL (Craig, 1961).

## 3   Model evaluation results

The following section present first results and comparisons of model simulations with measurements spanning several spatio-temporal scales: Sec. 3.1 shows how the models capability to simulate diagnostic $H_2O$ can be used to investigate source regions of the modeled water cycle. Sec. 3.2 then compares results for precipitation from a long model integration with measurements taken from the GNIP network (Terzer et al., 2013; IAEA/WMO, 2017). Sec. 3.3 looks at seasonal and regional differences by comparing model output with pairs of $\{H_2O, \delta D\}$ derived from IASI satellite measurements (Schneider et al., 2016). Finally, Sec. 3.4 presents a first case study, in which simulated values of $\delta D$ are compared with measurements from the CARIBIC project (Brenninkmeijer et al., 2007).

The following sections use modeled values of $H_2O$ (ICON standard water) and of HDO and $H_2^{18}O$. For the simulations discussed in the following sections, the diffusion constant ratio is set to the values of Merlivat and Jouzel (1979) and the equilibrium fractionation is parameterized following Majoube (1971) over liquid water and following Merlivat and Nief (1967) over ice. Surface evaporation is described by the parameterization by Pfahl and Wernli (2009). The parameterization by Merlivat and Jouzel (1979) has little influence on the values in the free troposphere and is not further discussed. Grid scale evaporation of hydrometeors is described by the parameterization by Stewart (1975) if not noted differently.

In addition to ICON standard water, three diagnostic water species are simulated. All fractionation is turned off, so they resemble $H_2O$. But the evaporation and initialization is different: water indexed by `init` (e.g. $q^{\text{init}}$) is set to ICON water at initialization, but evaporation is turned off. In the course of the simulation, the water of this type precipitates out of the model atmosphere. Water indexed by `ocn` and `lnd` on the contrary are initialized with zero and evaporate from the ocean ($q^{\text{ocn}}$) and land areas ($q^{\text{lnd}}$) respectively. The sum of $q^{\text{init}}$, $q^{\text{ocn}}$ and $q^{\text{lnd}}$ always equals the mass mixing ratio of ICON standard water, indexed as $q^{\text{ICON}}$.

### 3.1   An application of dignostic water tracers: Precipitation source regions

This section examines the source regions of vapor that is removed again by precipitation over ocean and land, respectively. The study uses two model runs that were initialized on November 5, 2013 and May 5, 2014 and both simulated four months. Sea surface temperature and sea ice cover were updated daily, by linearly interpolating monthly data provided by the AMIP II

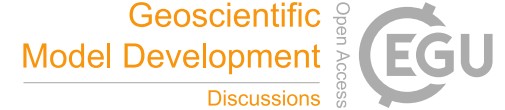



project (Taylor et al., 2000). A horizontal resolution of R2B06 was used, which corresponds to approximately $40\,\mathrm{km}$ (Zängl et al., 2015) and the time step was set to $180\,\mathrm{s}$, with the convection timestep called every second time step.

This section looks at the total precipitation $P$ in January and July for the two model runs initialized in November and May, respectively. Figure 1 displays zonal sums of $P^{\mathrm{init}}$, $P^{\mathrm{ocn}}$ and $P^{\mathrm{lnd}}$ relative to standard water precipitation $P^{\mathrm{ICON}}$ as a function of latitude. The sum of precipitation that orginiates from convection is also given for each water species.

The area covered by ocean is not equally distributed over different latitudes, which is the reason why ocean and land points are considered separately. The center panels show the fraction of precipitation that has fallen over the ocean relative to the total precipitation and the area fraction of the ocean in each latitudinal band. Despite the different characteristics of the different seasons, which will be discussed in the following paragraphs, the latitudinal distribution of the ocean area fraction largely determines the overall fraction of rain that falls over the ocean or over land. This is why the other panels display values of $P$ relative to the sum over each compartment, not to the total sum.

Figure 1 shows the precipitation two months after initialization. At this time, the tropospheric moisture has almost completely been replaced almost completely by water that has evaporated during the model run. This is demonstrated by the very low values of $P^{\mathrm{init}}$ in all four panels. Technically, this means that the ternary solution of $q^{\mathrm{init}}$, $q^{\mathrm{ocean}}$ and $q^{\mathrm{land}}$ that makes up $q^{\mathrm{ICON}}$ is practically reduced to a binary solution of only $q^{\mathrm{ocean}}$ and $q^{\mathrm{land}}$. $q^{\mathrm{init}}$ does not have to be considered in the troposphere two months after initialization.

During northern hemisphere winter over the ocean (top left panel of Fig. 1), the precipitation is strongly dominated by water that has evaporated from the ocean. Water from the land surface hardly reaches the ocean. Over land areas, the ocean is also the dominant source for precipitation, reaching more than 50% at almost all latitudes. In the northern and southern hemisphere mid latitudes, more than 70% of the precipitated water originates from the ocean. Only the tropics receive up to 40% of precipitation from land evaporation. Most precipitation at tropical and subtropical latitudes over the ocean originates from convection (indicated by dashed lines), while the role of convection is much smaller over land areas. This process depends on model resolution. In a simulation with very high horizontal resolution, more convective processes could have been directly resolved. The amount of precipitation from convection therefore points at the importance of this parameterization in the model.

The situation is different in northern hemisphere summer (bottom row of Fig. 1), especially in the northern hemisphere. In summer, the northern hemisphere land areas (bottom right) supply themselves a substantial fraction of the moisture that then precipitates. The importance of convection is slightly increased in northern hemisphere summer with its maximum influence shifted northward. Despite the larger moisture availability over the ocean, the northern hemisphere land areas also supply the larger part of moisture that precipitates over the ocean in July.

This first application of ICON-ART-Iso - while no isotopologues are used - shows how diagnostic moisture tracers can be applied to better understand specific aspects of the atmospheric water cycle.

## 3.2 A multi annual simulation compared to GNIP data

For a first validation of $\delta\mathrm{D}$ and $\delta^{18}\mathrm{O}$ values, this section uses a multi annual ICON-ART-Iso model integration and compares results to data taken from the GNIP network (Global Network for Isotopes in Precipitation, see Terzer et al., 2013;





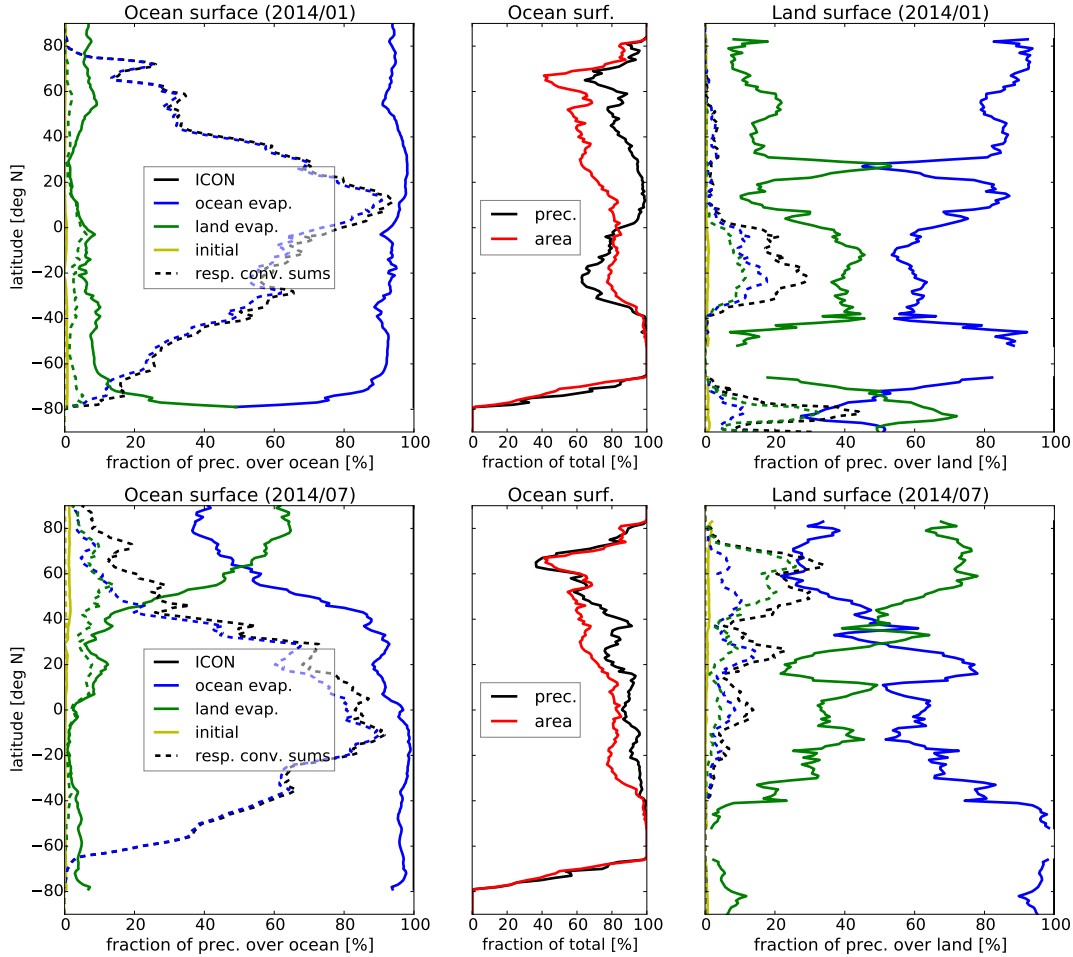

**Figure 1.** Fractional contributions of $P^{ICON}$, $P^{init}$, $P^{ocn}$ and $P^{lnd}$ to zonal sums of total precipitation for January (top) and July (bottom), as a function of latitude. Left and right panels show sums over the ocean and land grid points, respectively. Dashed lines indicate the contribution of convective precipitation for each source of atmospheric water. Center panels display the fraction of precipitation over the ocean relative to total precipitation (over land plus ocean) and the fraction of the area covered by ocean.





IAEA/WMO, 2017). The simulation was initialized with ECMWF (European Centre for Medium-Range Weather Forecast) Integrated Forecast System (IFS) data at January 1, 2007, 0UTC, to simulate almost 7 years on an R2B04 grid ($\approx 160\,\mathrm{km}$ horizontal resolution). The time step was set to $240\,\mathrm{s}$ (convection called every second step) and output was saved on a regular $1° \times 1°$ grid every 73 h in order to obtain values from all times of day. Sea surface temperatures and sea ice cover are, again,

updated daily, using the data provided by Taylor et al. (2000). The first 91.25 days are not considered as spin-up time of the model.

Five GNIP stations were chosen for their good data coverage in the respective years, sampling different climate zones: Karlsruhe in southwestern Germany ($49.0°N$,$8.3°E$), Vienna in eastern Austria ($48.2°N$,$16.3°E$) in central Europe, Ankara in central Anatolia ($40.0°N$,$32.9°E$), Puerto Montt in central Chile ($41.5°S$,$72.9°W$) and Halley station in Antarctica ($75.6°S$,$20.6°W$).

The closest grid point to each of these stations was taken from the model output and the multi-year mean was calculated for each calender month of $\delta^{18}$O and d-excess in precipitation, total precipitation $P$ and two meter temperature $T_{2m}$. The corresponding values are available from the GNIP database. Results are displayed in Fig. 2. The panels for total precipitation also include the mean values of precipitation from ocean and land evaporation (see previous section). d-excess values have been calculated as d-excess $= \delta\mathrm{D} - 8\,\delta^{18}$O. All panels also show the intervals of the $1\sigma$ standard deviation for model and measurement

data.

For most stations, the seasonal cycle of precipitation is reproduced by the model. This includes the summer minimum for Ankara and the strong winter precipitation in Puerto Montt. Precipitation is overestimated for Karlsruhe by roughly 50%. For all stations, the influence of land evaporation is strongest in their respective summer. The central European stations and Ankara show a decreasing influence of the ocean in winter with increasing distance to the ocean and a more continental climate. For

Puerto Montt, located between the Pacific and the Andean mountain range, and for Halley station, almost all precipitating water originates from the ocean.

The seasonal cycle of temperature is reproduced for all stations. Winter temperatures are too cold in the model for all stations. This temperature bias can partly be explained by the fact that the altitude of all stations is higher in the model because of the coarse grid, e.g. $550\,\mathrm{m}$ for the grid point identified with Vienna versus $198\,\mathrm{m}$ for the GNIP station. Also, the

measured temperatures are slightly higher than mean monthly ERA-Interim (Dee et al., 2011) two meter temperatures for the corresponding grid points (not shown).

Considering these meteorological biases, it can be stated that mean values of $\delta^{18}$O are well reproduced by ICON-ART-Iso for all five stations. The seasonal cycle is captured correctly on the northern as well as on the southern hemisphere. Values of d-excess are also of correct magnitude. Model data is more variable than the measurements, but they are often in the standard

deviation range of measurements. This demonstrates the capability of ICON-ART-Iso to simulate climatological patterns. The seasonal cycle and regional differences in $\delta\mathrm{D}$ and $\delta^{18}$O are correctly reproduced by the model in multi-year mean values of precipitation.





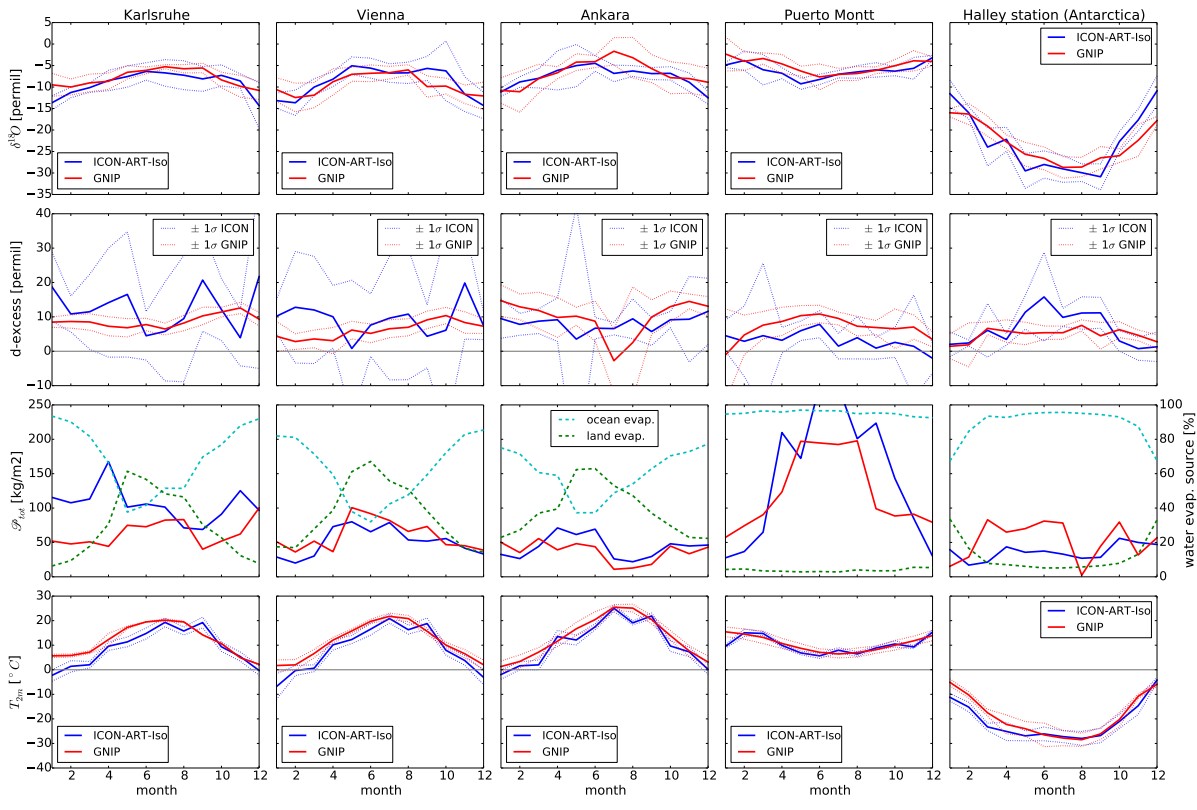

**Figure 2.** Monthly mean data for five GNIP stations (left to right: Karlsruhe, Vienna, Ankara, Puerto Montl and Halley station). Variables listed from top to bottom: $\delta^{18}$O, d-excess ($\delta$D $- 8\,\delta^{18}$O), total precipitation $P_{\mathrm{tot}}$ and two meter temperature ($T_{2m}$). Plots showing $P_{\mathrm{tot}}$ also include the percentage of land and ocean evaporation in precipitation. All figures (except $P$ for sake of readability) also indicate the one standard deviation ($\sigma$) interval (dotted lines).

### 3.3 Comparison with IASI satellite data for a seasonal perspective

This section compares pairs of $\{H_2O, \delta D\}$ retrieved from MetOp/IASI remote sensing measurements with data from two simulations. In doing so, the section closely follows the case studies presented by Schneider et al. (2017), who compare IASI and data from the global, hydrostatic model ECHAM5-wiso (Werner et al., 2011).

5 ### 3.3.1 IASI satellite data and model postprocessing

IASI (Infrared Atmospheric Sounding Interferometer) A and B are instruments on board the MetOp-A and MetOp-B satellites (Schneider et al., 2016). They measure thermal infrared spectra in nadir view from which free tropospheric $\{H_2O, \delta D\}$ pair data are derived. As the satellites circle the earth in polar, sunsynchronous orbit, each IASI instrument takes measurements



twice a day at local morning (approximately 9:30) and evening (approximately 21:30) hours. These data are most sensitive in approximately $4.9\,\mathrm{km}$. An IASI $\{H_2O, \delta D\}$ pair retrieval method has been developed and validated in the framework of the project MUSICA (MUlti-platform remote Sensing of Isotopologues for investigating the Cycle of Atmospheric water). The MUSICA retrieval method is presented by Schneider and Hase (2011) and Wiegele et al. (2014) with updates given in

Schneider et al. (2016).

Schneider et al. (2017) present guidelines for comparing model data to the remote sensing data. First, a Retrieval Simulator software is used for simulating the MUSICA averaging kernel, using the atmospheric state of the model atmosphere. The simulated kernel is than applied to the original model state ($\boldsymbol{x}$) in order to calculate the state that would be reported by the satellite retrieval product ($\hat{\boldsymbol{x}}$, see Eq. 23).

$$\hat{\boldsymbol{x}} = \boldsymbol{A}(\boldsymbol{x} - \boldsymbol{x}_\mathrm{a}) + \boldsymbol{x}_\mathrm{a} \qquad (23)$$

Here, $\boldsymbol{A}$ is the simulated averaging kernel and $\boldsymbol{x}_\mathrm{a}$ the a priori state. The a priori value used in the retrieval process for $4.9\,\mathrm{km}$ is at $\{1780\,\mathrm{ppm}, -217.4\,\mathrm{permil}\}$. This value represents the climatological state of the atmosphere. In the retrieval process, the satellite radiance measurements are used for estimating the deviation of the actual atmospheric state from the a priori assumed state, where it is important to note that the remote sensing retrieval product is not independent from the a priori assumptions

(see Schneider et al. (2016) for more details). In Schneider et al. (2017), these guidelines have been followed for comparison of IASI data with ECHAM5-wiso model data. We use the same approach for comparisons to ICON-ART-Iso and our results can be directly compared to the results achieved with the hydrostatic, global model ECHAM5-wiso.

In order to compare ICON-ART-Iso measurements with IASI data, two simulations of four months are used, which were initialized on November 5, 2013, and May 5, 2014 (the same simulations have been used in Sec. 3.1). As in Schneider et al.

(2017), two target time periods are investigated from February 12-18 and August 12-18, respectively. As has been shown in Sec. 3.1, the amount of water remaining in the troposphere from initialization is negligible by using lead times of three months (also for vapor, not shown). For this study, model output was interpolated to a regular $0.36° \times 0.36°$ grid, which is close to the $40\,\mathrm{km}$ (R2B06) resolution of the numerical ICON grid close to the equator. Output was written for every hour of simulation.

IASI observations are only available at cloud free conditions. In order to exclude cloud affected grid points in the ICON

data, the total cloud cover simulated by ICON was used, denoted by $C_\mathrm{clct}$. All points with $C_\mathrm{clct} > 90\%$ were excluded. Surface emissivity $E_\mathrm{srf}$ is a necessary input parameter for the Retrieval Simulator. In this first study, $E_\mathrm{srf}$ was set to 0.96 over land and 0.975 over the ocean. This is in accordance with the mean values as given by Seemann et al. (2008). In addition, Schneider et al. (2017) show in a sensitivity study that errors on the order of 10% in this value have only a limited influence on the averaging kernels as simulated by the Retrieval Simulator. From the output of the Retrieval Simulator, values were only used where the

sensitivity metric $s_\mathrm{err} < 0.05$, as recommended by Schneider et al. (2017). This assures meaningful results.

The scatter of $\{H_2O, \delta D\}$ is not shown directly. Instead, the figures show the isolines of relative normalized frequency, which is explained in App. A. In addition, Rayleigh fractionation curves are indicated in all figures. These are the same as those given by Schneider et al. (2017).



### 3.3.2 Seasonal and daily cycle

Seasonal and daily cycle are investigated in $\{H_2O, \delta D\}$ space. The seasonal cycle is discussed for different regions over the central Pacific Ocean. The daily cycle is considered in the tropics and subtropics, also investigating differences between land and ocean areas.



**Figure 3.** Isolines of the relative normalized frequency distribution for pairs of $\delta D$ and $H_2O$ (see App. A for this method) after processing ICON-ART-Iso data with the IASI Retrieval Simulator of Schneider et al. (2017) (top) and IASI data for the same time (bottom). Data from morning overpasses are shown for 12 to 18 February (left) and 12 to 18 August (right), 2014 for different latitudinal bands over the Pacific Ocean (longitude $\lambda < 140°W$ or $\lambda > 140°E$). Contour lines are indicated at 0.2, 0.6 and 0.9 of the distribution.

5    First, the seasonal cycle over the Pacific Ocean is examined by comparing the two target periods in different areas ($\lambda < 140°W$ or $\lambda > 140°E$ longitude and different latitudinal bins). Results are presented in Fig. 3, which also gives the exact





latitudes. IASI data (bottom panels) show specific characteristics of the different regions. $H_2O$ content is highest for tropical air masses and lowest for the highest latitudes in February and August. At the same time, tropical air is least depleted in HDO, while the highest latitudes show the lowest values of $\delta D$, i.e. are more depleted. When comparing February and August values at each latitude, a clear seasonal signal appears everywhere except for the tropics: During summer of the corresponding

hemisphere, the air is more humid and more depleted in HDO. The distributions seem to shift from season to season along a line perpendicular to those of the Rayleigh model. The distribution in the tropics shows a broadened shape in August.

The results of ICON-ART-Iso are shown in the top panels of Fig. 3. The latitudinal dependence is similar to IASI: high $H_2O$ and $\delta D$ in the tropics and the lower values for mid-latitudes. The range of values is also very similar. The seasonal cycle in $H_2O$ and $\delta D$ is also reproduced to some degree, especially in the subtropical latitudes. The most obvious differences occur in

the northern hemisphere mid-latitudes in summer, which show less negative values of $\delta D$ in the model than in the satellite data, especially for humid situations. In winter, this may also be the case, but there are only few humid values simulated at all. In general, the model shows a similar behavior as ECHAM5-wiso, the results of which are presented by Schneider et al. (2017).

For the daily cycle in the tropics and subtropics, land and ocean points are considered separately (Fig. 4, see caption for exact definition of the bins). IASI shows a clear signal of the daily cycle for both the tropics and subtropics over land (bottom

panels of Fig. 4). There is no such signal over the ocean, where morning and evening distributions are almost identical. Over land, the water vapor in the tropics and subtropics is more depleted of HDO in the morning. There is also a daily cycle in $H_2O$ in the tropics: During morning overpasses, $H_2O$ values are higher than in the evening. Schneider et al. (2017) argue that this is due to the cloud filter, which removes areas of heavy convection in the evening. In the morning, the clouds have disappeared, but high humidity remains, especially in the lower troposphere. This is partly due to evaporation of rain drops, which explains

the enhanced depletion in HDO (Worden et al., 2007). Over the Sahara (the subtropical land area considered), the daily cycle is different: While mixing ratios of $H_2O$ rise only slightly during the day, there is a strong increase in the HDO content in the evening. This behavior can be attributed to vertical mixing (Schneider et al., 2017, and references therein).

The data retrieved from ICON-ART-Iso model simulations is shown in the top panels of Fig. 4. Tropical air (top left) over the land shows slightly lower mixing ratios for $H_2O$ than IASI. The humidity of tropical ocean points is better reproduced. The

difference in $\delta D$ is stronger for both areas, with $\delta D$ values being too high in the model. There is no daily cycle in the tropics for ICON-ART-Iso. The subtropical mixing ratios (top right) of $H_2O$ over the ocean are similar to those in the tropics but cover a smaller range than those retrieved from IASI. The very humid and very dry parts of the IASI distribution are not reproduced by the model. Over land, the values retrieved from ICON-ART-Iso show a weak daily cycle, which is, however, much smaller compared to the IASI data. $\delta D$ values in ICON-ART-Iso are substantially larger than in the IASI retrievals. As pointed out by

Schneider et al. (2017), the daily cycle in IASI also manifests itself in the number of samples passing the IASI cloud filter and quality control. The IASI cloud filter removes much more evening observations than morning observations, meaning more cloud coverage in the evening than in the morning. In contrast, the ICON-ART-Iso cloud filter removes a similar number of data for morning and evening simulations, i.e. in the model morning and evening cloud coverage is rather similar. This may also influence the results.







**Figure 4.** Isolines of the relative normalized frequency distribution for pairs of $\delta$D and $H_2O$ (see App. A for this method) after processing ICON-ART-Iso data with the IASI Retrieval Simulator of Schneider et al. (2017) (top) and IASI data for the same time (bottom). Left: Data corresponding to morning and evening overpasses for the tropics ($10^\circ S < \varphi < 10^\circ N$, all longitudes, summer and winter simulation) over land and over the ocean. Right: Morning and evening overpasses for the subtropics ($22.5^\circ S < \varphi < 35^\circ N$, summer simulation) over land (Saharan desert region, $10^\circ W < \varphi < 50^\circ E$) and Atlantic Ocean ($50^\circ W < \varphi < 30^\circ W$). Contour lines are indicated at 0.2, 0.6 and 0.9 of the distribution.





To further analyze the influence of ocean and land areas, the analysis of the daily cycle is repeated, making use of the humidity tracers $q^{ocn}$ and $q^{lnd}$. As has been shown in Sec. 3.1, $q^{init}$ is negligible already two months after initializations. To distinguish between grid points mostly influenced by ocean or land evaporation, we additionally use the following criteria: $q^{ocn}/q^{ICON} > 0.9$ for oceanic grid points, $q^{lnd}/q^{ICON} > 0.5$ for land grid points predominantly affected by land evaporation.

The contributions of water evaporating from ocean and land have not been processed with the retrieval simulator, instead values interpolated to $4.9\,\text{km}$ are directly used.

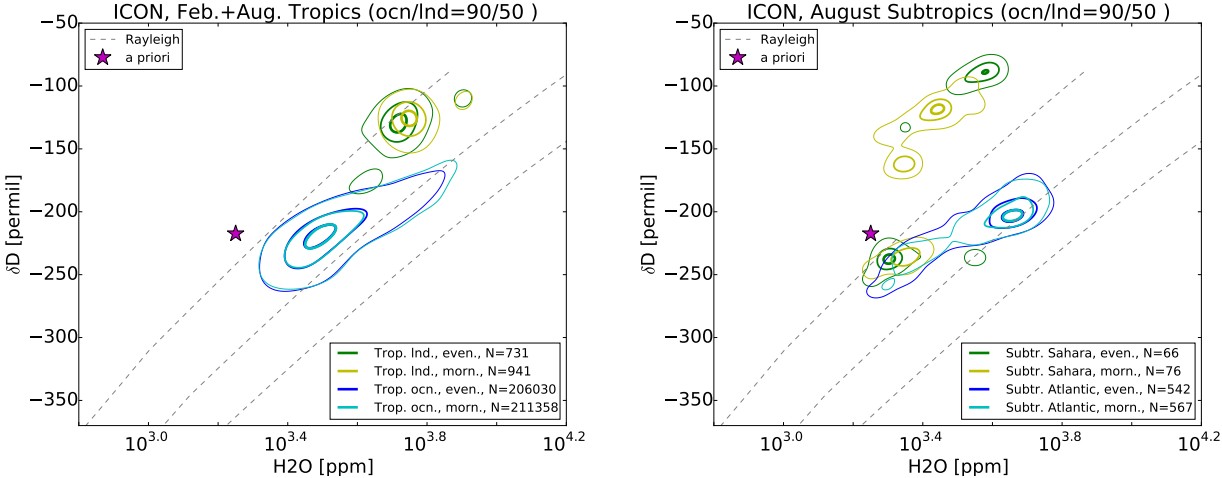

**Figure 5.** As Fig. 4 for ICON-ART-Iso. In addition to the land-ocean mask, land data must pass the condition $q_v^{lnd}/q_v > 0.5$ and ocean data must pass $q_v^{ocn}/q_v > 0.9$.

The result is shown in Fig. 5 for tropics and subtropics, using the same method as for Fig. 4. The characteristics of the different regions show up much more clearly with these additional criteria. For the tropical ocean, the distribution of $H_2O$ is similar, but the values are slightly more depleted in HDO, removing that end of the distribution that apparently has a strong

influence by land evaporate. The distribution of pairs attributed to the land surface is reduced to values with relatively high humidity and enriched in HDO. The latter might be due to the signal of plant evapotranspiration, which is considered a non-fractionation process.

In the subtropics, the distributions over land change their shape completely and are partly separated from those over the ocean. The distribution for the subtropical ocean remains largely unchanged, becoming slightly more elongated with lower

values in $\delta D$. For the land surface, the additional criaterion strongly reduces the number of values that are considered. 50% land evaporate over the Saharan desert implies two different regimes: Either the air is very dry and highly processed (low $\delta D$), or the moisture content is large with relatively high values in $\delta D$. This subset of the full distribution with high $\delta D$ values also shows a strong daily cycle which is on the same order of magnitude as in the IASI data of Fig. 4.



This section shows that ICON-ART-Iso is able to reproduce regional differences and the seasonal cycle of $\{H_2O, \delta D\}$ of vapor in the lower troposphere. The additional water diagnostics are used to study the behavior of the model in more detail and may help in investigating the measured distributions in future studies.

### 3.4 Comparing with in situ IAGOS-CARIBIC measurements

This section presents a first case study, in which results of ICON-ART-Iso are compared to in situ measurements of $\delta D$ taken by the IAGOS-CARIBIC passenger aircraft at 9-12 km altitude. Two flights in September 2010 are considered, which took place a few days after the passage of the tropical cyclone Igor. The full dataset of all $\delta D$ measurements of IAGOS-CARIBIC taken in the tropics is also as reference.

### 3.4.1 IAGOS-CARIBIC data and model postprocessing

In the European research infrastructure IAGOS-CARIBIC, a laboratory equipped with 15 instruments is deployed onboard a Lufthansa A340-600 for four intercontinental flights per month. Measurements of up to 100 trace gases and aerosol parameters are taken in situ and in air samples (Brenninkmeijer et al., 2007). $\delta D$ is measured using the instrument ISOWAT (Dyroff et al., 2010). It is a tunable diode-laser absorption spectrometer that simultaneously measures HDO and $H_2O$ at wave numbers near $3765 \, cm^{-1}$ to derive $\delta D$ in vapor. The instrument is calibrated based on regular calibration measurements (each 30 min) of
a water vapor standard with 500 ppm $H_2O$ and $\delta D = -109$ permil. The $\delta D$ offset is derived by considering the data of the driest 5% of the air masses sampled during each flight, which is typically 4-8 ppm $H_2O$. At the flight altitude of 10-12 km, this is without exception lowermost stratospheric air (LMS), for which a $\delta D$ of -600 permil is assumed (Pollock et al., 1980; Randel et al., 2012). An assumed uncertainty of this LMS value of 400 permil translates to a relevant uncertainty of 20 permil at 100 ppm $H_2O$. Due to further sources of measurement uncertainty, the data has a total flight specific systematic uncertainty
up to 100 permil. The total uncertainty is humidity dependent, decreasing towards higher humidity (e.g. 100 permil at 80 ppm $H_2O$ vs. less than 20 permil at 500 ppm $H_2O$, see Christner (2015) for more details).

  The in-situ IAGOS-CARIBIC data is suitable for the analysis of processes on small scales. $\delta D$ measurements are available as mean over one minute, which translates to a spatial scale of approximately 15 km. This horizontal resolution exceeds the chosen ICON-ART-Iso configuration and is therefore very suitable for validation in form of a case study. Unfortunately, the
uncertainty of $\delta D$ data at humidity below approximately 40 ppm $H_2O$ is too high to be used for analysis. Because of the systematic total uncertainty (see above), we use mean $\delta D$ values from two flights through similar conditions.

  In this section, measurements from a return flight from Frankfurt to Caracas on September 22, 2010 are analyzed (IAGOS-CARIBIC flight nrs. 309 and 310). The two flights crossed the Atlantic approximately two days after Hurricane Igor passed the flight track. The storm caused large-scale lofting of tropospheric air masses and a moistening at flight level. The high humidity
at flight level (9-12 km) allowed many accurate $\delta D$-measurements to be taken.

  An ICON-ART-Iso simulation was initialized with ECMWF IFS analysis data from September 12, 2010 and with the isotope values as explained in Sec. 2.7. This corresponds to a ten day forecast for the time of the two flights. Not all tropospheric water from initialization has been replaced by water evaporated during the simulation at this time. But the $\delta D$ values adjust to local



meteorology within a few days, developing realistic horizontal and vertical gradients. The simulation was set up on an R2B06 grid ($\approx 40\,\mathrm{km}$) with a time step of $240\,\mathrm{s}$ (convection called every second step). The hourly output was examined on a $0.5°$ regular grid and interpolated linearly to the position of the aircraft. Fig. 6 shows $\delta$D in water vapor in the upper troposphere along with the flight paths of the IAGOS-CARIBIC flights. The flights cross the Atlantic shortly after the hurricane, sampling the model atmosphere where it has been influenced by the storm.

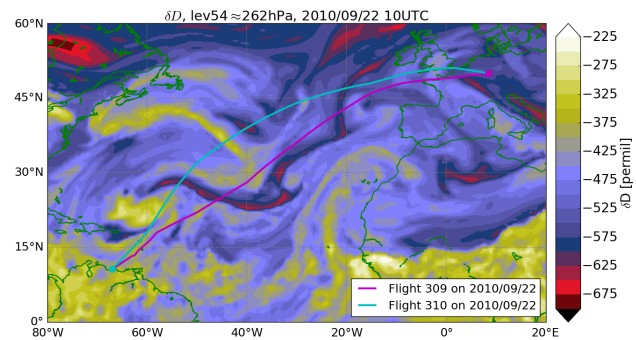

**Figure 6.** $\delta$D in water vapor on model level 54 ($\approx 260\,\mathrm{hPa}$) on September 22, 10UTC, the date of the IAGOS-CARIBIC flights. The storm is visible in the western half of the plotted area (center approximately at $20°N, 60°W$). The flight paths of IAGOS-CARIBIC flights 309 and 310 are also indicated by two lines, where the departure location is emphasized.

### 3.4.2 Results for flights in tropical regions

In order to compare data processed by hurricane Igor, model and measurement data from flights 309 and 310 are considered in latitudes around the storm track only ($0°N < \varphi < 30°N$). A sample of tropical $\delta$D values taken from a model simulation and the IAGOS-CARIBIC database is also examined for reference. As in Sec. 3.3, the distribution of pairs of $\{H_2O, \delta D\}$ are

examined. The results are shown in Fig. 7.

The distribution of IAGOS-CARIBIC $\delta$D-measurements is shown in the top left panel of Fig. 7 (panel A). The tropical measurement sample (blue contours) consists of all respective measurements ($23.5°S < \varphi < 23.5°N$). While most tropical values are centered around -500 permil in $\delta$D and 100 ppm $H_2O$, there is also a tail towards more humid pairs in the distribution. The lower limit in $\delta$D follows the curves of Rayleigh fractionation. The measurement data from flights 309 and 310 (red contours)

show different characteristics. The range in $H_2O$ is similar to the maximum density values of all tropical values, but the samples are more depleted in HDO. The humid branch is missing. In general, both distributions are limited by the detection limit of 40 ppm in $H_2O$, while contour lines may reach slightly lower values because of the smoothing that is applied in processing the data (see App. A).

To create the larger sample of pairs of $\delta$D and $H_2O$ from model data, a longer simulation of ICON-ART-Iso was used (the

same simulation that was compared with IASI data in August 2014, see Sec. 3.3). 30 output files were randomly chosen and





from each file and 200 data points were randomly drawn from tropical latitudes in each file. Data was only considered within the pressure range of $180\,\mathrm{hPa} < p < 280\,\mathrm{hPa}$ to consider the flight altitude of IAGOS-CARIBIC. Together with the samples along the paths of flights 309-310, the probability distribution are shown in the right panels of Fig. 7 (panels B and D). For the top panel (B), the parameterization of Stewart (1975) for fractionation during evaporation of the hydrometeors is used (as in all

other sections). For comparison, the lower right panel (D) uses the parameterization of Blossey et al. (2010). Model data laong the flight tracks are used only where accurate deld-measurements are available.

A limit of $40\,\mathrm{ppm}$ is also applied to the model data. The isolines covering lower value pairs again result from smoothing the data. The bottom left panel of Fig. 7 (panel C) shows the distributions without this limit (using the parameterization of Blossey et al., 2010). The distribution is different for dry situations, which shows that there are situations which are not captured by the

IAGOS-CARIBIC measurements.

The distribution from the tropical model sample is similar to the one by all tropical CARIBIC measurements (comparing the blue contours of panel A to B and D): There is a tail towards high humidities and the upper limit of $\delta$D is roughly at -400 $\mathrm{permil}$, while the lower limit is given by the second Rayleigh curve. This is true for both parameterizations of fractionation during evaporation.

The distribution for the two flights is slightly different between IAGOS-CARIBIC and ICON-ART-Iso(comparing the red contours of panel A to B and D). The modeled distribution includes more humid values than the measurements. It is shifted to more negative values in $\delta$D relative to the full tropical sample, as in the measurements. From this simulation and these measurements alone, it is difficult to say if these discrepancies result from errors in the meteorological representation of the hurricane or in the physical parameterizations of the model. The good agreement between model and measurements in general

is promising, while details will need to be examined in a future study.

In this respect it is interesting to compare the two parameterizations for fractionation during evaporation (panels B and D of Fig. 7). The parameterization of Blossey et al. (2010), panel D, produces values of $\delta$D that are approximately 50 $\mathrm{permil}$ lower than those using the parameterization by Stewart (1975), panel B. At the first impression, the results using the parameterization by Blossey et al. (2010) seem more similar to the IAGOS-CARIBIC measurements, but it is again difficult to decide from this

model run alone. The comparison demonstrates the capability of the model to simulate different realizations of HDO during one model integration. It also shows the range of influence that a specific parameterization can have on the results.

By using the other three diagnostic moisture tracers (initialization water $q^{\mathrm{init}}$ and water evaporating from the ocean and land, $q^{\mathrm{ocn}}$ and $q^{\mathrm{lnd}}$), the model results are examined further. The two transatlantic flights spent little time over land areas. Accordingly, $q_v^{\mathrm{lnd}}$ only reaches an average of 3-5% for both flights, 15% at maximum. Roughly 50% of the sampled water originates from

the initialization, while the remainder has evaporated from the ocean in the course of the simulation. Part of the discrepancies between model and measurements may thus result from the rough representation of $\delta$D in the initial vapor field.

This is analyzed in more detail in Fig. 8. Values of $\delta$D and $H_2O$ along the flight paths are combined with information on the origin of the water that is sampled in the model. Here, $W$ stands for the ratio of vapor that originates from land or ocean evaporation or initialization, e.g. $W^{\mathrm{init}} = q_v^{\mathrm{init}}/q_v$. Because $W^{\mathrm{lnd}}$ is very low during the longer parts of the flights,



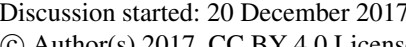



**Figure 7.** Isolines of the relative normalized frequency distribution (contours at 0.1, 0.4 and 0.9, see App. A) of IAGOS-CARIBIC measurements (top left, A) and ICON-ART-Iso model simulations for tropical samples and IAGOS-CARIBIC flights 309-310. Model data is interpolated onto the paths of the IAGOS-CARIBIC flights for two realizations of HDO using different parameterizations of fractionation during evaporation of hydrometeors (following Stewart (1975), top right, panel B, and Blossey et al. (2010), bottom right, panel D). Model data is considered only in locations where there are also measurements and the measurement limit of $40\,\mathrm{ppm}$ is considered (right panels B and D). In the bottom left panel C, this limitation is dropped, the parameterization of Blossey et al. (2010) is used here. The tropical model samples were created by randomly drawing data from a four months simulation (of Sec. 3.1), considering the limits in pressure of IAGOS-CARIBIC ($180\,\mathrm{hPa} < p < 280\,\mathrm{hPa}$).




$W^{\mathrm{ocn}} = 1 - W^{\mathrm{init}}$ is a good approximation. In Fig. 8, green colors indicate $W^{\mathrm{lnd}} > 10\%$, where the approximation of a binary solution is not valid.

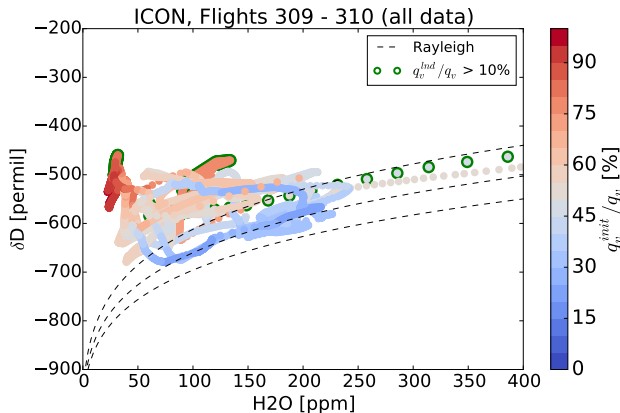

**Figure 8.** Scatter of $\delta$D against $H_2O$ from ICON-ART-Iso interpolated to the flight paths of IAGOS-CARIBIC flights 309 and 310, using the parameterization of Stewart (1975). Color coding indicates the ratio of $W^{\mathrm{init}} \approx 1 - W^{\mathrm{ocn}}$ in percent. Locations with $W^{\mathrm{lnd}} > 10\%$ are marked in green. Values are considered where $p < 280\,\mathrm{hPa}$.

Fig. 8 shows the strong influence of the ocean. More than 50% of the sampled vapor originates from ocean evaporation for long parts of the flights. Water with high ratios of $q_v^{\mathrm{lnd}}$ is generally less depleted in $H_2O$.

Those values with the highest values of $W^{\mathrm{init}}$ are very dry and show show high $\delta$D. Vertical cross sections along the flight tracks (not shown) indicate that these values come from the second half of flight 310, which encountered an intrusion of upper tropospheric or stratospheric air. This part of the model atmosphere is strongly influenced by the initialization profile. The lowest values in $\delta$D are reached where $W^{\mathrm{ocn}}$ is high (low values of $W^{\mathrm{init}}$) and the air has been heavily processed by the model.

The pairs of $\{H_2O, \delta D\}$ displayed in Fig. 8 show signals of different processes, which can be read from the scatter (see

Worden et al., 2007). Only short parts along of the flight tracks sample data indicating a Rayleigh process. A broad range of values is simulated for high $W^{\mathrm{ocn}}$ or high $W^{\mathrm{init}}$. This demonstrates that ICON-ART-Iso is capable of capturing different non-Rayleigh processes. This includes air mass mixing, but also the microphysical and convective processes that are imprinted on the isotopologue ratio.

## 4   Conclusions

We present ICON-ART-Iso, the isotope enabled version of the global atmospheric model ICON. The article describes the model formulation of as well as a set of evaluation studies. By using parts of the ICON-ART infrastructure, the model is very flexible in terms of the simulated moisture tracers. These can be set to resemble either $H_2O$ (tagged water) or the stable isotopologues HDO or $H_2^{18}O$ if fractionation is turned on. The physics of fractionation are largely based on the model COSMOiso. The





first part of this article gives a detailed explanation of the parameterizations that have been implemented in ICON-ART-Iso to simulate the fractionation of water isotopologues.

We first evaluate $H_2O$ tracers of moisture evaporating from land and ocean to investigate the moisture sources of precipitation. This demonstrates the capabilities of ICON-ART-Iso to use tagged water as an additional diagnostic. The following three
sections then investigate the performance of the model for the simulation of the isotopologues, considering (i) multi annual, (ii) regional and (iii) meso-scale applications.

For a multi annual evaluation, the simulated isotopologues HDO and $H_2{}^{18}O$ from a seven year simulation on a relatively coarse grid (160 km horizontal resolution) are compared to measurements taken from the network of GNIP stations. The model is shown to simulate $\delta D$ and $\delta^{18}O$ reasonably well, reproducing the seasonal cycle of $\delta^{18}O$ and the range in d-excess
for different stations in the northern and southern hemisphere. The analysis demonstrates the long term stability of the model and presents a first climatological application.

Regional differences in pairs of $\{H_2O, \delta D\}$ in lower free-tropospheric water vapor are then compared to data retrieved from IASI satellite measurements for a summer and winter case. The latitudinal dependence of these pairs is comparable to those from IASI retrievals. The seasonal cycle over the Pacific ocean and the overall values are reproduced by the model in both
seasons. The difference between land and ocean surfaces in the tropics and subtropics in the model is of similar magnitude as in the measurements. However, the daily cycle that is observed in the satellite data is not reproduced in the model. Overall, the performance is similar to that of ECHAM5-wiso (see Schneider et al., 2017).

In a meso-scale application, a first comparison with in situ measurements uses $\delta D$ in upper-tropospheric water vapor from two IAGOS-CARIBIC flights transecting the Atlantic and from all tropical IAGOS-CARIBIC measurements. ICON-ART-Iso
is able to reproduce the general features of the tropical IAGOS-CARIBIC dataset. The characteristics of the samples taken during two flights shortly after Hurricane Igor in September 2010 are also captured by the model. This study is used to discuss differences that appear in the results because of different parameterizations for fractionation during the evaporation of hydrometeors.

In all three applications, the tagged evaporation water from the ocean or land surfaces proofs to be a valuable tool. It reveals
a seasonal cycle in the precipitation water origin or shows the influence of the initialization in case of the comparison with IAGOS-CARIBIC data.

ICON-ART-Iso is a promising tool for future investigations of the atmospheric water cycle. This study demonstrates the flexibility of the model in terms of the setup for different diagnostics but also in terms of horizontal resolution and time scale. For future applications, it will be interesting to implement a nudging of meteological variables towards analysis data to facilitate
comparisons with measurements in case studies. Fractionation will be implemented in different microphysical schemes to make the model numerically more efficient and even better applicable to climatological questions. Due to its flexible setup, ICON-ART-Iso is ready to simulate instances of $H_2{}^{17}O$ or to be used as a testbed for new microphysical parameterizations.

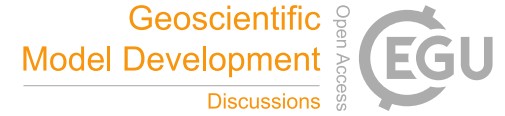

*Code and data availability.* The CARIBIC measurement data analyzed in this paper can be accessed by signing the CARIBIC data protocol to be downloaded at http://www.caribic-atmospheric.com/. The ICON code can be obtained from DWD after signing the license agreement available from icon@dwd.de. The ICON-ART code can be obtained after signing the license agreement available from bernhard.vogel@kit.edu.

## 5  Appendix A:  Preparing the relative normalized frequency distributions

Sec. 3.3 and 3.4 discuss distributions of $\{H_2O, \delta D\}$. The scatter of $\{H_2O, \delta D\}$ is not shown directly as the figures would be too cluttered. Instead, the normalized relative frequency is discussed, isolines of which are shown in the different figures. This method has been adopted from Christner (2015). Fig. A1 shows the scatter and the isolines of normalized relative frequency for the IAGOS-CARIBIC measurements of flights 309-310, which are discussed in Sec. 3.4.

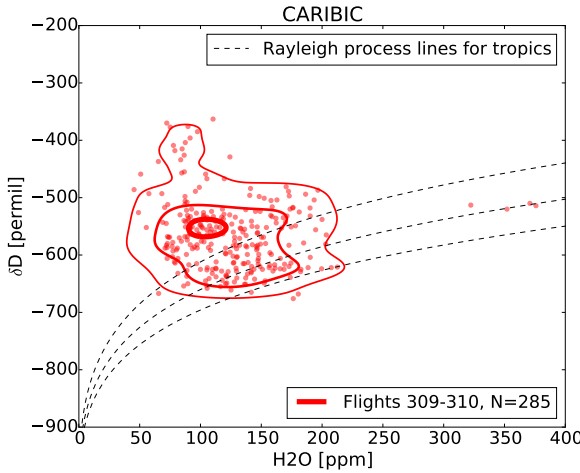

**Figure A1.** Scatter and isolines of the relative normalized frequency distribution for tropical (latitude $\varphi < 30°N$) measurements of $\delta D$ and $H_2O$ from IAGOS-CARIBIC flights 309 and 310 (September 2010). The figure demonstrates how the isolines (indicated at 0.1, 0.4 and 0.9) relate to the underlying scatter.

10    To arrive at the isolines, the data is binned in $H_2O$ and $\delta D$ on a grid of $5\,\mathrm{ppm} \times 5\,\mathrm{permil}$. In case of IASI data, the data is binned in $\log_{10} H_2O$ (ppm) $\times \delta D$ on a grid of $0.05 \times 5$. Histogram counts are then interpolated onto a $1000 \times 1000$ grid. This is smoothed with a Gaussian filter with a standard deviation of 20 (15 in case of IASI). This smoothed data is then normalized by the sum of all value pairs and then normalized by the maximum value. Within this array of smoothed counts, isolines are drawn at 0.9, 0.4 and 0.1 (0.9, 0.6 and 0.2 in case of IASI).





*Author contributions.* Johannes Eckstein programmed ICON-ART-Iso as an extension to ICON, performed and evaluated the simulations and prepared the manuscript. This was done with Roland Ruhnke as main and Peter Braesicke as as overall advisor. Stephan Pfahl provided the code of COSMOiso and helped in understanding and implementing the fractionating code. Daniel Reinert was the main contact person at DWD and helped with the model ICON. Daniel Rieger and Jennifer Schröter aided in the technical development of the ICON-ART part

of the model code and by helping to solve many technical problems during the development of ICON-ART-Iso. Emanuel Christner provided and discussed data and results of the comparison with IAGOS-CARIBIC. Andreas Zahn is the coordinator of IAGOS-CARIBIiC. Christoph Dyroff was responsible for setting up and maintaining the instrument ISOWAT. Matthias Schneider provided data and discussed results of the comparison with IASI satellite data.

*Competing interests.* There are no competing interests to declare.

*Acknowledgements.* The authors would like to thank A. Seifert and M. Raschendorfer of DWD for discussions and help with parts of the ICON model code. This work was partly performed on the computational resource ForHLR II funded by the Ministry of Science, Research and the Arts Baden-Württemberg and DFG ("Deutsche Forschungsgemeinschaft"). The MUSICA/IASI data have been produced in the framework of the projects MUSICA (funded by the European Research Council under the European Community's Seventh Framework Programme (FP7/2007-2013) / ERC Grant agreement number 256961) and MOTIV (funded by the the Deutsche Forschungsgemeinschaft

under GZ SCHN 1126/2-1). We thank all the members of the IAGOS-CARIBIC team. The collaboration with Lufthansa and Lufthansa Technik and the financial support from the German Ministry for Education and Science (grant 01LK1301C) are gratefully acknowledged.



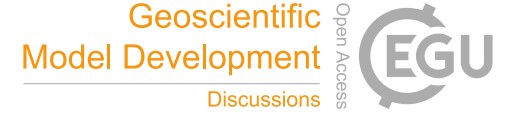

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
