# Peer review of "From climatological to small scale applications: Simulating water isotopologues with ICON-ART-Iso (version 2.1)"

_Geoscientific Model Development, 2017_

## Referee Comment (RC1) · Anonymous Referee #1 · 25 Jan 2018

Review of "From climatological to small scale applications: Simulating water isotopologues with ICON-ART-Iso (version 2.1)", submitted to GMD by Eckstein et al. Paper number GMD-2017-280

This papers describes the implementation of water isotopologues and water tracers into ICON, a new global non-hydrostatic model, along with an assessment of initial results from the model.

Assessment:

The paper does a good job of describing ICON and the implementation of water tracers and isotopologues into the model and its physical parameterizations. The valida-

tion studies move from surface precipitation to mid-tropospheric vapor to upper tropospheric vapor and provide a good picture of how the model captures isotopic variations. I suggest below that the different schemes for isotopic exchange between rain and vapor might be better evaluated in a situation where measurements of deuterium excess are available and also in the lower troposphere where most rain evaporation occurs.

Recommendation: Minor revisions.

Major comment:

1. Why perform the sensitivity study on the parameterizations for isotopic exchange during evaporation from rain on water vapor in the upper troposphere? Also, since rain evaporation predominantly happens in the lower troposphere, I would argue that this sensitivity study be performed either for precipitation or for near-surface/lower-tropospheric vapor. Since rain evaporation is a non-equilibrium process, I would expect differing parameterizations to have the strongest impact on deuterium excess rather than deltaD or delta18O.

In this paper, measurements of deuterium excess are only available for the GNIP data, so that this would argue for the evaporation test to be applied on that data. Alternately, a recent dataset of isoptopic measurements in water vapor (Benetti et al, 2017, Scientific Data, https://doi.org/10.1038/sdata.2016.128 ) might provide a reasonable constraint on isotopic values over the ocean surface. I would suggest that the authors try out the evaporation scheme sensitivity study for the GNIP data and possibly also for this vapor dataset (perhaps using climatological monthly values for the appropriate locations). If they help differentiate between the performance of the two evaporation parameterizations, I would suggest that the CARIBIC comparison only apply to the standard model and the evaporation test applied to lower-tropospheric vapor or precipitation.

Minor comments (5/26 means page 5, line 26):

2/30: I think that this statement should be qualified as "isotopologue enabled _global_

models", since COSMOiso (Pfahl et al, 2012) and SAM (Blossey et al, 2010) are non-hydrostatic.

3/15-20: Might it be useful to specify a few sample grid resolutions here for the different applications of ICON?

5/eqn 2: I find the choice of alpha_eq < 1 to be surprising, but if this is carried consistently throughout the code, I suppose it fine. If one looks at Majoube (1971), for example, one finds log(alpha_18O) = 1.137e3/Tˆ2 - 0.4156/T - 2.0667e-3, which if my computations are correct yields 0.0117. This suggests alpha itself is greater than one. Still, as I said, if this interpretation of alpha is applied consistently through the code, that seems okay, but be careful to comment things well so that the next coder who is accustomed to alpha > 1 is not confused. I would suggest that the variable might be renamed to something other than alpha to avoid confusion in the future. Checking these equations against those in Blossey and Stewart required a lot of careful attention, with the alpha switching meaning between those papers and this one.

6/eqn 6: First, I am a bit worried by equation 6 with the saturation vapor deficit in the denominator since this will go to zero in the cloud. Of course, the evaporation rate of the standard isotope will also go to zero, but careful coding is required when dividing small numbers by similarly small numbers. Secondly, this equation seems to shut off isotopic exchange between rain and vapor in saturated conditions. This doesn't seem to happen in COSMOiso as described by Pfahl et al (2012, equations 4-5). Perhaps, the implementation in ICON preserves isotopic exchange in saturated conditions. If so, this should be mentioned in the text.

6/eqn 7: While this expression seems consistent with equation 7 from Blossey et al, the denominator is needlessly obscure. The quantity in parenthesis in the denominator can be written as $(1 + b_l)/ (\hat{l}D \rho_{l,\infty}^*)$. I would advocate putting the $(1+b_l)$ in the denominator and putting the $\hat{l}D$ next to the f and the $\rho_{l,\infty}^*$ at the end.

Also, my impression from looking at both Stewart and Blossey is that they are working

from the same basic equations. I'm guessing that if the formula for ˆl S_xˆ{evap} were plugged into equation 6, something very close to equation 7 would emerge, though with the (hD/lD)ˆn in place of the combination of the diffusivity ratio and ventilation factor ratio. There is probably a way of writing these two equations so that they are easy to compare by eye and don't require a lot of algebra.

Last, note that alpha>1 in equation 7 and alpha<1 in equation 6, unless my algebra was wrong.

7/eqn 10: Shouldn't that be ˆlf/ˆhf in the denominator? Maybe eta could be defined as the product of the diffusivity and ventilation factor ratios: (ˆhD/ˆlD) * (ˆhf/ˆlf) to simplify this formula and also equation 7.

8/6-7: Could the sentence "Evaporation of precipitation ..." be rephrased? The effects of evaporation on buoyancy and the resulting cold pools is certainly important, but it is important for other reasons as well. Thinking of isotopic applications, Risi et al (2008, JGR, doi:10.1029/2008JD009943) suggest that the recycling to vapor from downdrafts into the boundary layer could play a role in the amount effect. As this vapor is affected by rain evaporation and re-equilibration in the downdraft, evaporation could be an important process for this isotopic application as well.

9/sec 3: Are these simulations free-running or are they nudged to reanalysis fields (wind, temperature, surface pressure) to preserve the "observed" meteorology? It's worth making this clear, because isotope-enabled global models are often run in nudged configurations to produce a sort of isotope reanalysis (e.g., Steen-Larsen et al, 2017, JGR, http://dx.doi.org/10.1002/2016JD025443 ).

13/fig 2: I realize that Vienna is the home of the IAEA, but does it really provide much information that isn't in the Karlsruhe plot? Is it worth having both here when they're so similar? Removing one location might enable the panels to be larger and more readable, which seems desirable.

14/25: I think that the threshold for cloud-affected grid points should perhaps be set much lower. I would advocate for 50% at most and think of 10% as a better characterization of the almost-cloud-free conditions that would be ideal for a IASI retrieval.

16/19: Suggested re-wording "This _may be_ partly due to evaporation of rain drops ...". There could be other things contributing to these changes in the 4-6km layer where IASI is most sensitive, so that I would suggest less certainty here. I think of rain evaporation as being most prominent below cloud base, less so in the mid-troposphere, though I am happy to be corrected on this.

Typographic suggestions:

1/6: "... measurements _of_ precipitation ..."

1/12: "... as well as _that of all_ tropical data ..."

2/1: "It is the strongest green house gas (...) _and_ distributes energy through the release of latent (...), while liquid and frozen particles influence the radiative balance (...)." I don't think "to name only three ..." is needed.

2/13-15: Possible rephrasing: "Measurements of the isotopic content of vapor first required cryogenic samplers (Dansgaard, ...), but in the last 15 years laser absorption spectroscopy has made in situ measurements possible (citations)."

6/6: "ilead" —> "lead"

9/2: "presents"

9/19: comma before "respectively".

10/2: Remove "timestep" after "convection". It's unnecessary.

10/15: Try to avoid starting a sentence with a symbol when possible. Suggested rephrasing: "Two months after initialization, q_{init} ..."

10/24: "... importance of this parameterization in the _simulations at this resolution._"

In other simulations with the same model or at other resolutions, this parameterization might not have the same role.

10/25: To make this flow better, suggested rephrasing: "The situation is different in northern hemisphere summer (...). Land areas in the northern hemisphere (bottom right) themselves supply a substantial fraction ..."

12/11: Move d-excess definition here: "... and d-excess (= deltaD - 8 delta18O) in precipitation." and remove the sentence on lines 13-14.

14/2: "in approximately" –> "at a height of approximately"

20/20: "Thirty output files ..." At the start of a sentence, spell out the number (or rephrase to avoid having it at the start).

21/1: "... from each file, and 200 ..." This is a compound sentence, so that there should be a comma before the "and".

21/2: "to consider" –> "corresponding to"

21/2-3: "The resulting probability distributions are shown in the right panels of Fig. 7 (...) together with the samples along the paths of flights 309-310." I think this is more clear for the reader.

24/24: "proofs" –> "proves"

---

## Referee Comment (RC2) · Anonymous Referee #2 · 30 Jan 2018

This paper presents the new isotope-enabled model ICON-ART-Iso. It describes the main equations used for the fractionation processes and presents first evaluation results compared to various datasets and at different scales.As such, it deserves publication and it is well suited for GMD. For the evaluation, it makes use of some very recent observations, which is an additional strength of this paper.

My main regret is that he model-observations comparison could be more quantitative, and the physical processes responsible for the model-observations mismatches could be more discussed. But this is maybe not the priority for this first paper. Therefore, I recommend acceptation with minor revisions.

[Figure]

**Detailed comments**

- p2 l 1: isotope-enabled

- p2 l 19: it's awkward here to contrast "climatological questions" and "process understanding". Climatological questions can be answered through process understanding. In addition, limited area models are not the only tool to understand processes, global models can also be used for this purpose. Reformulate, by highlighting rather the differences in spatial scales or in convective representation.

- p 3 l1: "precipitation diagnostics" is mysterious here -> precipitation source regions?

- p 3 l 19: "climate prediction" -> "climate projections? It's impossible to predict climate for the end of the century. In case you refer to studies at the decadal scale, climate predictability is more appropriate.

- p 4 l 15: guarantee

- p4 l 17: syntax problem

- p 5 l 21: no, Risi et al 2010 and Werner et al 2011 did not use such a simple assumption. In stand-alone mode, both LMDZ and ECHAM models use a bucket model that collects the precipitation to represent the soil reservoir. To my knowledge, you are the first to make such a simple assumption. It's not a problem but it should be mentionned.

- p 8 l 5: for the liquid fraction, does it mean that you assume that all drops are sufficiently small to equilibrate totally? If so, it's not a problem but it should be mentionned that this is a simplifying assumption compared to offline models of rain-vapor exchanges in saturated environment (e.g. Stewart (1975); Lee and

Fung (2008) calculate the equilibration as a function of the drop size) or to GCM parameterizations (e.g. Hoffmann et al. (1998) assume that only a proportion of the drops equilibrate in a saturated environment depending on the precipitation type).

- p 8 l 30: how do you initialize water vapor composition at the model bottom and top? Interpolation needs these end members in addition to the tropopause values.

- p 9 l 9-14: why is this paragraph here and not in the Methods section?

- p 10 l 12: remove one "almost completely"

- p 10, section 3.1: previous studies using water tagging in models or water tracking tools should be cited and their results could be briefly compared to yours: e.g. Joussaume et al. (1984); Koster et al. (1986); Numaguti (1999); Yoshimura et al. (2004); van der Ent et al. (2010); Gimeno et al. (2012); Risi et al. (2013)

- p 18 l 13-18: but this does not improve the model-observations agreement...

- p 18: can the lack of daily cycle be related to the wrong precipitation daily cycle, a known problem in many models (Betts and Jakob (2002); Guichard et al. (2004))?

- p 20 l 9: how is this "sample" chosen?

- p 21 l 22: what explains the $\delta D$ difference between these 2 parameterizations? Explain with simple physics what process is the main driver of this change. In addition, I understand that these 2 parameterizations differ for the representation of isotopic processes during rain evaporation, which occurs in the lower troposphere. Why does it have such a big impact in the upper troposphere?

- p 22 fig 7. A few explanations on these distributions could be useful. For example, do you see the signature of condensate lofting/detrainement? In the upper

tropospere, these processes are known to have a big impact on the isotopic composition (Moyer et al. (1996); Kuang et al. (2003); Bony et al. (2008); Sayres et al. (2010)). This could be discussed.

- p 23 l 9: this is very vague: how can different processes be identified from a scatter? Be more precise.

- p 23 l 10: remove "of"

- Overall section 3.4.2 and previous sections: be more quantitative when describing the model-observations agreement. Use quantitative metrics such as RMS error. This would allow to compare quantatively the model-observations agreement between different model version, different regions and seasons, different sampling criteria... It's difficult to assess the model-observations agreement by comparing by eye 2 different plots.

- p 23 l 5-8: is the model-observations agreement better when removing these parcels with high proportion of initialization tracer?

- p 24 22: "discuss" -> show: you don't discuss the reasons for these differences.

- p 24 l 32: "instances" -> use a more appropriate word?

**References**

Betts, A. K. and Jakob, C. (2002). Study of diurnal cycle of convective precipitation over Amazonia using a single column model. *J. Geophys. Res.*, 107 (D23):4732–4745, doi:10.1029/2002JD002264.

Bony, S., Risi, C., and Vimeux, F. (2008). Influence of convective processes on the isotopic composition (deltaO18 and deltaD) of precipitation and water vapor in the Tropics. Part 1: Radiative-convective equilibrium and TOGA-COARE simulations. *J. Geophys. Res.*, 113:D19305, doi:10.1029/2008JD009942.

Gimeno, L., Stohl, A., Trigo, R. M., Dominguez, F., Yoshimura, K., Yu, L., Drumond, A. R. D. M., Duran-Quesada, A. M., and Nieto, R. (2012). Oceanic and terrestrial sources of continental precipitation. *Rev. Geophys.*, 50(4):doi:10.1029/2012RG000389.

Guichard, F., Petch, J., Redelsperger, J.-L., Bechtold, P., Chaboureau, J.-P., Cheinet, S., Grabowski, W., Grenier, H., Jones, C., Köhler, M., et al. (2004). Modelling the diurnal cycle of deep precipitating convection over land with cloud-resolving models and single-column models. *Quarterly Journal of the Royal Meteorological Society*, 130(604):3139–3172.

Hoffmann, G., Werner, M., and Heimann, M. (1998). Water isotope module of the ECHAM atmospheric general circulation model: A study on timescales from days to several years. *J. Geophys. Res.*, 103:16871–16896.

Joussaume, S., Jouzel, J., and Sadourny, R. (1984). A general circulation model of water isotope cycles in the atmosphere. *Nature*, 311:24–29.

Koster, R., Jouzel, J., Suozzo, R., Russell, G., Broecker, W., Rind, D., and Eagleson, P. (1986). Global sources of local precipitation as determined by the NASA/GISS GCM. *Geophy. Res. Lett.*, 13 (2):121–124, DOI:10.1029/GL013i002p00121.

Kuang, Z., Toon, G., Wennberg, P., and Yung, Y. L. (2003). Measured HDO/H2O ratios across the tropical tropopause. *Geophysical Research Letters*, 30:25–1.

Lee, J.-E. and Fung, I. (2008). "Amount effect" of water isotopes and quantitative analysis of post-condensation processes. *Hydrological Processes*, 22 (1):1–8.

Moyer, E. J., Irion, F. W., Yung, Y. L., and Gunson, M. R. (1996). ATMOS stratospheric deuterated water and implications for troposphere-stratosphere transport. *Geophys. Res. Lett.*, 23:2385–2388.

Numaguti, A. (1999). Origin and recycling processes of precipitating water over the Eurasian continent: Experiments using an atmospheric general circulation model . *J. Geophys. Res*, 104:D2, 1957–1972, doi:10.1029/1998JD200026.

Risi, C., Noone, D., Frankenberg, C., and Worden, J. (2013). Role of continental recycling in intraseasonal variations of continental moisture as deduced from model simulations and water vapor isotopic measurements. *Water Resour. Res.*, 49:4136–4156, doi: 10.1002/wrcr.20312.

Sayres, D. S., Pfister, L., Hanisco, T. F., Moyer, E. J., Smith, J. B., Clair, J. M. S., O'Brien, A. S., Witinski, M. F., Legg, M., and Anderson, J. G. (2010). Influence of convection on the water isotopic composition of the tropical tropopause layer and tropical stratosphere,. *J. Geophys. Res.*, 11:D00J20, doi:10.1029/2009JD013100.

Stewart, M. K. (1975). Stable isotope fractionation due to evaporation and isotopic exchange

of falling waterdrops: Applications to atmospheric processes and evaporation of lakes. *J. Geophys. Res.*, 80:1133–1146.

van der Ent, R. J., Savenje, H. H. G., Schaefli, B., and Steele-Dunne, S. C. (2010). Origin and fate of atmopheric moisture over continents. *Water Resour. Res.*, 46:W09525.

Yoshimura, K., Oki, T., Ohte, N., and Kanae, S. (2004). Colored moisture analysis estimates of variations in 1998 asian monsoon water sources. *J. Meteor. Soc. Japan*, 82:1315–1329.

---

## Referee Comment (RC3) · Anonymous Referee #3 · 1 Feb 2018

In their paper, Eckstein, J. et al. present a general circulation model that has now been equipped with the water isotopologues HDO and $H_2^{18}O$. It comprises a model description and an evaluation with respect to precipitation, satellite data and in-situ aircraft measurements. Moreover, a "tagged water" concept is presented that can be used for process studies.

Essentially, the paper presents an endeavour that had been conducted numerous times before, a lot of (more than 10) GCMs comprise water isotopologues. Recently, technical advances have been made with respect to advancing and extending the implemen-

tation of water isotopologues in GCMs or ESMs, e.g. Xu et al. 2012 have integrated the isotopologues in a coupled global ocean model, Haese et al. 2013 have included water isotoplogues to a coupled atmosphere-land surface model, Eichinger et al. 2015 have added stratospheric methane oxidation effects. This paper here, however, does unfortunately not include anything novel. Also the evaluation with respect to precipitation and to satellite measurements had been conducted before with other models, in part even in extensive model intercomparison projects (SWING). There are also mistakes in the model conception (surface), but see below at my major comments.

The usage of CARIBIC data for comparison with an isotope enabled GCM is indeed new and interesting, this part of the paper unfortunately is very disappointing. First, it is not well structered, it seems like the authors want to compare everything at once here, instead of doing it point by point and thus no clear message shines through. Second, the evaluation should be much more quantitative and several points should be discussed in more detail here. And third, the method of the model-data comparison with respect to the simulation setup is not suitable for the given goal of the study. See more details on this in the major comments section.

The tagged water concept is a nice tool for process studies, but the concept a) is simple and does not need a reference in GMD, b) has also been implemented many times before and c) is sort of detached from the rest of the paper and d) does not provide any important insigths here. More details in my major points.

Moreover, the manuscript requires a general polishing of language and formulations, it is written in a somewhat halting manner with errors and inaccuracies. I provide some examples below in the minor comments and technical sections.

Altogether, I do not think that this paper adds considerable value to water isotope model development. In my opinion, a work that only repeats what had been done before with yet another model, should rather find its place in a short method section, or in the supplement of the paper of another journal, along with an in-depth research study

using the new tool and answering the one or the other science question. Anyway, although the paper is really disappointing, I guess it is still valid to publish such an article in GMD just to have a reference (if that is what GMD wants), but at least the points I am making below should be considered before publication.

**Major comments:**

- There are two serious flaws in Section 2.3 that have to be corrected for to achieve a state-of-the-art water isotope enabled GCM:

  – The authors are mistaken when they think that $R_{surf}$ is approximated by $R_{VSMOW}$ in Werner et al., 2011. In ECHAM5-wiso, the surface reservoirs soil, skin, snow and plant layer are filled by precipitation and form $R_{surf}$, which is then used for evaporation processes from the surface. Only fine processes like isotope fractionation during evapotranspiration are neglected. Using $R_{VSMOW}$ for the entire surface is a step backwards in global water isotope modelling.

  – Also as a lower boundary condition for the ocean, this is not a state-of-the-art approximation and it is not done this way in recent models. Instead, a global gridded data set based on the $^{18}$O isotopic composition in sea water by LeGrande and Schmidt, 2006 (see attached figure) is taken e.g. in Werner et al., 2011. HDO in the ocean surface layer can be approximated from this.

- P6L5-6: Wrong. In fact, during freezing of water isotopic effects occur in a closed systems (Souchez and Jouzel, 1984) and there is even evidence of a kinetic effect (Souchez et al. 2000), but due to the comparably low diffusivities in liquid water these effects can be neglected in cloud processes in GCMs. This has to be mentioned somewhere, such inaccuracies should not make their way into scientific literature.

- Fig. 2 and Sect.3.2: You forgot to include a station in the tropics. Other processes (amount effect,...) become important here that have to be evaluated.

- The simulation setup and the initialization procedure is not suitable for the given goal of the study. To have a one-to-one comparison of water isotopologues in model results and observations a free running simulation with such short spin-up from such a crude initialisation is not applicable, but let me make these points one by one:

  1. Since your aim is to evaluate your representation of water isotopologues (including the water cycle processes as well as fractionation effects) in the model and not the model meteorology, you first have to compare a nudged (specified dynamics) simulation. Otherwise you will never be able to separate the effects of an unequal meteorological situation with the water and isotope effects. And that is what you need to do in the first step, because only then you can really assess your water isotope implementation. Or the other way around, your evaluation is pointless with the used simulation.

  2. In particular with such a crude initialization of HDO, you need longer spin-up time, maybe a month or more. Then you will not face problems like in P21L31, so why not just do it? However, probably already the HDO field of your first simulation could serve as a (somewhat better) initialisation field for your second simulation.

- As mentioned above, the section on the comparison with CARIBIC data should be comprehensively restructured. Comparisons should be made point by point and clear conclusions should be drawn and presented out of that.

- This brings me to my next point. The authors correctly write, that water isotope modelling is applied to answer climatological questions and (this should be "and" not "or" on page 2, see other REF) process understanding. This usually means the simulation of a particular phenomenon for in-depth analysis with the additional

information from the water isotopes. Here, the authors are using the tool in a weather prediction setup, but they never clarify why they chose this configuration, i.e. how one could profit from this sort of "weather model setup". Or in other words, what advantages does this setup have over the application of the isotopes in a regular climate model configuration.

- The concept of the tagged water tool is simple and it has been developed many times before (which the authors do not even mention), see REF#2. Thus, the concept itself does not require another reference in GMD. The process studies that are conducted with it in this paper do not provide any new insights into the hydrological cycle. Plus, there is only little connection of it to the actual topic of the paper, the only real connection is that one can approximate the spin-up time with the approach. This, however, the authors will not need anymore if they consider my point on the simulation setup (which is crucial for the evaluation and thus for the entire paper). Hence, this part of the study dangles somewhere in the nowhere here and should thus be removed completely from the paper.

**Minor comments:**

- P1L6: This sentence needs to be rephrased, it is incorrect english and it is not clear how this shall represent a range of temporal scales.

- P2L7-9: This sentence is grammatically wrong (dangling participle).

- P2L26 and elsewhere: Cauquoin and Risi, 2017 has been rejected by GMD. You may want to find another citation for this.

- P2L31: Most (if not all) models can be run with a fine horizontal resolution. How does the ICON model "stand out" from other models with respect to this?

- P2L32: You use the word "flexible" here and also later on in the paper, and I think the word is being misused here. A model is not flexible when you can implement a lot of diagnostics, or run several water tracers at once. Basically you can do that with every model. Flexibility would be for example when you can easily switch these diagnostics and tracers on and off or change their attributes without having to recompile. Or being able to expand the model system such that you have several co-existing processes (e.g. convection schemes) that can be run with the same executable. I am not aware if this is possible in ICON-ART-Iso, if so you could describe it and use the word flexible, if not, you shJOURNAL OF GEOPHYSICAL RESEARCH, VOL. 110, D21314, doi:10.1029/2005JD005790, 2005ould rather use the word comprehensive or extensive.

- P2L32: You mention "diagnostic moisture tracers" and "tagged water" here, but at this stage the reader has no idea what you actually mean with that. Either explain it shortly, or wait until it comes to the point.

- P3L2 and L6 and throughout the manuscript: A section does not compare or discuss anything. These formulations make no sense, instead **you** compare it **in** the section. So use e.g.: In this section, we discuss....

- P3L26: And why is this a very good assumption? Because the abundance of standard water is at least three orders of magnitude higher than the abundance of any rare isotopologue. Why not include this in the text?

- P3L27-29: Rephrase: ...seven different forms (vapor, cloud water, ....), each of which is represented by one tracer for standard water and one tracer for the isotopologues HDO and $H_2^{18}O$, respectively.

- P4L1: What settings?

- P4L21-22: Sentence is unclear, rephrase.

- P8L23-24: At this stage, it is totally unclear where the simulation of "tagged water" is supposed to aim at, so this sentence is rather confusing.

- P8L30: and throughout the manuscript: To avoid ambiguity, the $\delta$ should be complemented by the isotope, the molecule and if relevant also by the phase, such as: $\delta D(H_2O_{ice})$.

- Sect. 2.7: The text never says that you are using HDO measurments here (only in the table caption), that makes the read unnecessarily confusing

- P9L15-20: This paragraph leaves unclear what this is supposed to be good for, why do you do that?

- P9L22: Please rephrase, the sentence is unclear. Also, why are you doing that?

- Fig. 1: Give panels a,b,c,d,e,f and change text accordingly

- P12L4: 73h is a very coarse output time step. Why so coarse? That makes only 10 steps a month, I think that is too weak if you want a robust climatology.

- P12L5: Why 91.25? Magic number?

- P12L12-13: Shortly explain why you do that.

- P12L27-32: The temperature bias should have a large impact on the isotope ratios. That does not always seem to be the case, see .e.g. Ankara. How come? In the depiction it is hardly possible to see how far the values are off, the vertical axes are too coarse.

- P14L25: Why 90%? Another magic number?

- Sect. 3.3.1: Do you take the model output at the same times and the same locations as the satellite makes observations (overpasses)? For this comparison you should.

- Fig. 3 and 4: Give panels a,b,c,d and change text accordingly

- P18L4: Two more magic numbers, please explain.

- P19L16-17: That would mean the flights take place in the extratropics only, because in the tropics these altitudes are still the troposphere. The flights do however seem to cross the equator, something is wrong here!

- P19L26: What conditions? Weather in general, or humidity, or what?

- P20L4-5: The flights do not sample the model atmosphere, the flights take measurements of the real atmosphere. Your off-line interpolation of the model data samples the model atmosphere. Be more precise.
  Moreover, taking only 1h output for this comparison you should make clear that, with an approximate flight velocity of the aircraft of around 900km/h, you will only get less than 10 model points per flight you are interpolating your data to. That is very coarse given that you are aiming on comparing local phenomena. In fact, it would be better to use an on-line flight tracking tool with high temporal resolution for this.

- Sect 3.4.1: Please state when exactly was which flight.

- P20L7: The hurricane processed your data? Please rephrase.

- P20L8: What model simulation?

- P20L9: What exactly do you mean by "the IAGOS-CARABIC database is examined"?

- P21L1: I do not understand what you mean by "randomly drawn" and "randomly chosen", please explain. If you want to make a climatological comparison here, why not make a long simulation to have a fair comparison?

- P21L23-24: From this depiction I cannot even see this. You should make the evaluation much more quantitative.

- P21L31: This is one part of what I mean with my major point on the simulation setup. If you had a longer (some months) spin-up time, you would not have to face these problems.

- Fig. 7: What does the N stand for?

- P23L5-8: So if you want to discuss stratospheric intrusions you can not get around methane oxidation and its influence on $\delta$D(H$_2$O) in the stratosphere. Depending on from which altitude the air is transported downwards, this could mean strongly depleted or enriched water vapour. Also, work has already been done to parameterize this effect on HDO in GCMs (Schmidt et al. 2005, Eichinger et al. 2015), it is a shame that this is not even discussed here given that you are evaluating results in the UTLS.

- P23L8: "air has been processed by the model" What does that mean? Please rephrase.

- P23L9: "different processes" What processes? At least shortly list some.

- Sect. 4: Change title from "Conclusions" to "Summary (and outlook)", there are no conclusions that go further than what had already been written before.

- P24L10: What do you mean by "long term stability"?

**Technical issues:**

- P1L18: ...oceans are an unmatched ...

- P1L18: "a reservoir to dissolve" - refine wording

- P1L12-13: It is not clear what is meant by "all of tropical data" here.

- P2L8: ... isotopologues in **water** vapor ... - this inaccuracy appears several times

- P2L14-15: Refine the english. Suggestion: The isotopologue content of water vapor has first been measured by means of cryogenic samplers (Dansgaard, 1964). In the last 15 years, also laser absoption spectroscopy methods have been developed for that use.

- P3L7: ... with the results of ICON-ART-Iso simulations. (You compare the results, not the simulations)

- P3L14: ... is the water isotopologue enabled ...

- P3L25: To discriminate .... , unclear, rephrase.

- P3L26: For clarity, you should write $^1\mathrm{H}_2^{16}\mathrm{O}$ here.

- P4L1: Remove "As".

- P4L17 The parameterization that influence ...

- P4L29-30: The ratio of ratios?

- P4L31 (and throughout the section): Put equation 2 here, not at the end of the paragraph. Generally restructure the presentation of equations.

- P6L6: ilead

- P8L20: ... in the standard setup

- P8L23: initialization of the water isotopologues!

- P9L9-14: repetition

- P9L15 what are water species?

- P9L21 diagnostic

- P13L3: remove "In doing so"

- P18L15: criterion

- P19L8: ... in the tropics serves as reference.

- P19L23: ...resolution is finer than the ...

- P19L24: remove "very"

- P19L28: ... Hurricane Igor had passed ...

- P21L5: laong

- P21L6: deld

- P23L5: show show

- P23L10: ...along the ...

- P23L16: remove "of"

**References:**

Souchez et al. 2000, A kinetic isotope effect during ice formation by water freezing, GRL, 27:1923:1926

Souchez and Jouzel 1984, On the isotopic composition in $\delta$D and $\delta^{18}$O of water and ice. JoG, 30:369-372

Schmidt, G. A. et al. Modeling atmospheric stable water isotopes and the potential for constraining cloud processes and stratosphere-troposphere water exchange, JOURNAL OF GEOPHYSICAL RESEARCH, VOL. 110, D21314, doi:10.1029/2005JD005790, 2005

Haese, B. , Werner, M. and Lohmann, G. (2013) Stable water isotopes in the coupled atmosphere-land surface model ECHAM5-JSBACH , Geoscientific Model Development, 6 , pp. 1463-1480, doi:10.5194/gmd-6-1463-2013

Eichinger, R. , Jöckel, P. , Brinkop, S. , Werner, M. and Lossow, S. (2015) Simulation of the isotopic composition of stratospheric water vapour – Part 1: Description and evaluation of the EMAC model , Atmospheric Chemistry and Physics, 15 (10), pp. 5537-5555, doi:10.5194/acp-15-5537-2015

Xu, X. , Werner, M. , Butzin, M. and Lohmann, G. (2012) Water isotope variations in the global ocean model MPI-OM , Geoscientific Model Development, 5 , pp. 809-818 . doi:10.5194/gmd-5-809-2012

---

## Author Comment (AC1) · 26 Oct 2018

**Reply to Referee 1**

Comments on

> From climatological to small scale applications: Simulating water isotopologues with ICON-ART-Iso (version 2.1)

Under revision at Geosci. Model Dev. Discuss., https://doi.org/10.5194/gmd-2017-280

Dear referee,

thank you for taking the time to review the manuscript. Your exact reading and recalculation of the microphysics algebra have greatly improved the model and subsequently also the manuscript. The physics have been corrected and as a consequence, we redid all the simulations. After analyzing the results of the parameterization by Blossey, we decided to no longer compare the two parameterizations in the current manuscript. This is the biggest change to our original submission.

Below, you will find a point by point reply to all your comments. We have attached a manuscript highlighting all the differences between the original and the current version of the manuscript. This includes the changes from all three referees. For completeness, we have also attached the updated version of the manuscript.

On behalf of all coauthors,

Johannes Eckstein

**Major comment:**

1.

> *Why perform the sensitivity study on the parameterizations for isotopic exchange during evaporation from rain on water vapor in the upper troposphere? Also, since rain evaporation predominantly happens in the lower troposphere, I would argue that this sensitivity study be performed either for precipitation or for near-surface/lower-tropospheric vapor. Since rain evaporation is a non-equilibrium process, I would expect differing parameterizations to have the strongest impact on deuterium excess rather than deltaD or delta18O.*

> *In this paper, measurements of deuterium excess are only available for the GNIP data, so that this would argue for the evaporation test to be applied on that data. Alternately, a recent dataset of isoptopic measurements in water vapor (Benetti et al, 2017, Scientific Data, https://doi.org/10.1038/sdata.2016.128) might provide a reasonable constraint on isotopic values over the ocean surface. I would suggest that the authors try out the evaporation scheme sensitivity study for the GNIP data and possibly also for this vapor dataset (perhaps using climatological monthly values for the appropriate locations). If they help differentiate between the performance of the two evaporation parameterizations, I would suggest that the CARIBIC comparison only apply to the standard model and the evaporation test applied to lower-tropospheric vapor or precipitation.*

> As stated above, we no longer compare the two parameterizations in the manuscript. After analyzing results of the decadal simulation in comparison with the GNIP dataset, we decided that a discussion of the results would inflate the manuscript. Instead, we now focus solely on the parameterization by Stewart. The implementation of the parameterization by Blossey is noted and the physics are compared in a short appendix. The comparison in terms of results is postponed to a future study.

**Minor comments:**

2/30:

*I think that this statement should be qualified as "isotopologue enabled _global_models", since COSMOiso (Pfahl et al, 2012) and SAM (Blossey et al, 2010) are non-hydrostatic.*

As suggested, we have included "global".

3/15-20:

*Might it be useful to specify a few sample grid resolutions here for the different applications of ICON?*

Following your suggestion, we now state the resolution of DWD's operational forecast. The paragraph now ends with:
"At DWD, ICON is used operationally for global numerical weather prediction (currently 13 km horizontal resolution, with a nest of 6.5 km resolution), it already proved successful as a Large Eddy Simulation (LES) model (Heinze et al., 2017), and is currently prepared for climate prediction studies at MPI-M. More details on ICON are given by Zängl et al. (2015)."

5/eqn 2:

*I find the choice of alpha_eq < 1 to be surprising, but if this is carried consistently throughout the code, I suppose it fine. If one looks at Majoube (1971), for example, one finds log(alpha_18O) = 1.137e3/T^2 - 0.4156/T - 2.0667e-3, which if my computations are correct yields 0.0117. This suggests alpha itself is greater than one.*

*Still, as I said, if this interpretation of alpha is applied consistently through the code, that seems okay, but be careful to comment things well so that the next coder who is accustomed to alpha > 1 is not confused. I would suggest that the variable might be renamed to something other than alpha to avoid confusion in the future. Checking these equations against those in Blossey and Stewart required a lot of careful attention, with the alpha switching meaning between those papers and this one.*

We admit that your concern is justified. The definition of alpha was taken over from the code of COSMOiso. It is consistently used this way in the code and we therefore decided to also use it this way in the manuscript.

6/eqn 6:

*First, I am a bit worried by equation 6 with the saturation vapor deficit in the denominator since this will go to zero in the cloud Of course, the evaporation rate of the standard isotope will also go to zero, but careful coding is required when dividing small numbers by similarly small numbers.*

*Secondly, this equation seems to shut off isotopic exchange between rain and vapor in saturated conditions. This doesn't seem to happen in COSMOiso as described by Pfahl et al (2012, equations 4-5). Perhaps, the implementation in ICON preserves isotopic exchange in saturated conditions. If so, this should be mentioned in the text.*

Thank you for the exact reading of the manuscript and for making this point! The way we had previously implemented the model equations, isotopic exchange was only happening at subsaturated conditions. Fixing this mistake in the code was one of the corrections which required us to redo all simulations.

6/eqn 7:

*While this expression seems consistent with equation 7 from Blossey et al, the denominator is needlessly obscure. The quantity in parenthesis in the denominator can be written as (1 + b_l)/ (ˆlD\rho_{l,\infty}ˆ\* ). I would advocate putting the (1+b_l) in the denominator and putting the ˆlD next to the f and the \rho_{l,\infty}ˆ\* at the end.*

*Also, my impression from looking at both Stewart and Blossey is that they are working from the same basic equations. I'm guessing that if the formula for ˆl S_xˆ{evap} were plugged into equation 6, something very close to equation 7 would emerge, though with the (hD/lD)ˆn in place of the combination of the diffusivity ratio and ventilation factor ratio. There is probably a way of writing these two equations so that they are easy to compare by eye and don't require a lot of algebra. Last, note that alpha>1 in equation 7 and alpha<1 in equation 6, unless my algebra was wrong.*

Thank you for redoing the algebra. Following your suggestions, we have reformulated the two equations so that they can now be compared easily, which we show in the new appendix B. In addition, we have corrected alpha in the equation for Blossey et al. The reformulated equations now also show that they both turn into the same equation for standard water if the standard water variables are inserted instead of the heavy water variables. The equation resulting from this replacement and the subsequent simplifications is identical to the one used in the microphysical scheme of ICON. Note that we now present the comparison of the two equations in App. B and no longer in the main body text since we also do not discuss results of the parameterization by Blossey.

7/eqn 10:

*Shouldn't that be ˆlf/ˆhf in the denominator? Maybe eta could be defined as the product of the diffusivity and ventilation factor ratios: (ˆhD/ˆlD) \* (ˆhf/ˆlf) to simplify this formula and also equation 7.*

We have also corrected this mistake. It was only a typographical error in the manuscript.

8/6-7:

*Could the sentence "Evaporation of precipitation ..." be rephrased? The effects of evaporation on buoyancy and the resulting cold pools is certainly important, but it is important for other reasons as well. Thinking of isotopic applications, Risi et al (2008, JGR, doi:10.1029/2008JD009943) suggest that the recycling to vapor from downdrafts into the boundary layer could play a role in the amount effect. As this vapor is affected by rain evaporation and re-equilibration in the downdraft, evaporation could be an important process for this isotopic application as well.*

As suggested, the sentence has been changed to:
"Evaporation of precipitation below cloud base is an important process for several reasons: it leads to a drop in the temperature and therefore influences dynamics, but is also important for the isotopic composition (Risi et al., 2008)."

9/sec 3:

*Are these simulations free-running or are they nudged to reanalysis fields (wind, temperature, surface pressure) to preserve the "observed" meteorology? It's worth making this clear, because isotope-enabled global models are often run in nudged configurations to produce a sort of isotope reanalysis (e.g., Steen-Larsen et al, 2017, JGR, http://dx.doi.org/10.1002/2016JD025443 ).*

We now clearly state this at the end of the first paragraph: "All simulations discussed here are free-running."

13/fig 2:

*I realize that Vienna is the home of the IAEA, but does it really provide much information that isn't in the Karlsruhe plot? Is it worth having both here when they're so similar? Removing one location might enable the panels to be larger and more readable, which seems desirable.*

To account for a comment by Referee 3, we have decided to stick with 5 stations, but now include Bangkok as a tropical station instead of Karlsruhe.

14/25:

*I think that the threshold for cloud-affected grid points should perhaps be set much lower. I would advocate for 50% at most and think of 10% as a better characterization of the almost-cloud-free conditions that would be ideal for a IASI retrieval.*

The variable used for cloud cover in ICON is very strict in the sense that 100% are reached quickly. This is why 90% percent is a reasonable threshold.

16/19:

*Suggested re-wording "This _may be_ partly due to evaporation of rain drops ...". There could be other things contributing to these changes in the 4-6km layer where IASI is most sensitive, so that I would suggest less certainty here. I think of rain evaporation as being most prominent below cloud base, less so in the mid-troposphere, though I am happy to be corrected on this.*

As suggested, we have included the "may".

**Typographic suggestions:**

1/6: *"... measurements _of_ precipitation …"*

As suggested, we have changed "...measurements in precipitation…" to "...measurements of precipitation…"

1/12: *"... as well as _that of all_ tropical data …"*

Included "that of" as suggested.

2/1: *"It is the strongest green house gas (...) _and_ distributes energy through the release of latent (...), while liquid and frozen particles influence the radiative balance (...)." I don't think "to name only three ..." is needed.*

As suggested, the sentence now reads:
"It is the strongest green house gas (Schmidt et al., 2010) and distributes energy through the release of latent heat (Holton and Hakim, 2013), while liquid and frozen particles influence the radiative balance (Shine and Sinha, 1991)."

2/13-15: *Possible rephrasing: "Measurements of the isotopic content of vapor first required cryogenic samplers (Dansgaard, ...), but in the last 15 years laser absorption spectroscopy has made in situ measurements possible (citations)."*

We have changed the sentence according to your suggestion.

6/6: *"ilead" —> "lead"*

corrected as suggested

9/2: *"presents"*

The paragraph now starts with "In the following sections, we present first results and comparisons of model simulations..."

9/19: *comma before "respectively".*

Corrected as suggested

10/2: *Remove "timestep" after "convection". It's unnecessary.*

The paragraph has been removed due to restructuring the manuscript.

10/15: *Try to avoid starting a sentence with a symbol when possible. Suggested rephrasing: "Two months after initialization, q_{init} …"*

The sentence has been removed due to restructuring the manuscript.

10/24: *"... importance of this parameterization in the _simulations at this resolution._" In other simulations with the same model or at other resolutions, this parameterization might not have the same role.*

Similar to your suggestion, the paragraph now ends with:
"Still, it should be considered that the amount of precipitation from convection only shows the importance of this parameterization in the simulations at this resolution. At other resolutions, the parameterization might not have the same role."

10/25: *To make this flow better, suggested rephrasing: "The situation is different in northern hemisphere summer (...). Land areas in the northern hemisphere (bottom right) themselves supply a substantial fraction …"*

The paragraph has been restructured and now starts with:
"During northern hemisphere winter over the ocean (top left panel of Fig. 1), the precipitation is strongly dominated by water that has evaporated from the ocean. Water from the land surface hardly reaches the ocean."

12/11: *Move d-excess definition here: "... and d-excess (= deltaD - 8 delta18O) in precipitation." and remove the sentence on lines 13-14.*

changed as suggested

14/2: *"in approximately" –> "at a height of approximately"*

corrected as suggested

20/20: *"Thirty output files ..." At the start of a sentence, spell out the number (or rephrase to avoid having it at the start).*

This sentence has been removed as we now use the decadal simulation to generate the dataset.

21/1: *"... from each file, and 200 ..." This is a compound sentence, so that there should be a comma before the "and".*

This sentence has been removed as we now use the decadal simulation to generate this dataset

21/2: *"to consider" –> "corresponding to"*

The paragraph has been removed as we now use the decadal simulation.

21/2-3: *"The resulting probability distributions are shown in the right panels of Fig. 7 (...) together with the samples along the paths of flights 309-310." I think this is more clear for the reader.*

This paragraph has been removed as we now use the decadal simulation.

24/24: *"proofs" –> "proves"*

corrected as suggested

**Reply to Referee 2**

Comments on

From climatological to small scale applications: Simulating water isotopologues with ICON-ART-Iso (version 2.1)

Under revision at Geosci. Model Dev. Discuss., https://doi.org/10.5194/gmd-2017-280

Dear referee,

thank you for taking the time to review the manuscript. Your exact reading and especially the literature you have suggested have greatly improved the manuscript.

Below, you will find a point by point reply to all your comments. We have attached a manuscript highlighting all the differences between the original and the current version of the manuscript. This includes the changes from all three referees. For completeness, we have also attached the updated version of the manuscript.

On behalf of all coauthors,

Johannes Eckstein

**Minor comments:**

p2 l 1:

*isotope-enabled*

Following your suggestion, all occurrences of "isotope enabled" have been replaced by "isotope-enabled".

p2 l 19:

*It's awkward here to contrast "climatological questions" and "process understanding". Climatological questions can be answered through process understanding. In addition, limited area models are not the only tool to understand processes, global models can also be used for this purpose. Reformulate, by highlighting rather the differences in spatial scales or in convective representation.*

Following your suggestion, we have reformulated the sentence. It now reads:
"Many global and regional circulation models have been equipped to simulate the atmospheric isotopologue distribution, focusing on the global scale (Risi et al., 2010; Werner et al., 2011) or regional phenomena (Blossey et al., 2010; Pfahl et al., 2012, both limited area models)."

p 3 l1:

*"precipitation diagnostics" is mysterious here -> precipitation source regions?*

As suggested, the sentence now reads "Section 3.1 looks at precipitation source regions.".

p 3 l 19:

*"climate prediction" -> "climate projections? It's impossible to predict climate for the end of the century. In case you refer to studies at the decadal scale, climate predictability is more appropriate.*

As suggested, we have changed "climate prediction studies" to "climate projections".

p 4 l 15:

*guarantee*

corrected

p4 l 17:

*syntax problem*

Corrected: "The parameterizations which influence the water cycle also include processes that do not fractionate."

p 5 l 21:

*No, Risi et al 2010 and Werner et al 2011 did not use such a simple assumption. In stand-alone mode, both LMDZ and ECHAM models use a bucket model that collects the precipitation to represent the soil reservoir. To my knowledge, you are the first to make such a simple assumption. It's not a problem but it should be mentionned.*

Thank you for pointing this out. We have removed the sentence and restructured the paragraph, as we now use the dataset by LeGrande and Schmidt (2006) over the ocean surface.

p 8 l 5:

*For the liquid fraction, does it mean that you assume that all drops are sufficiently small to equilibrate totally? If so, it's not a problem but it should be mentionned that this is a simplifying assumption compared to offline models of rain-vapor exchanges in saturated environment (e.g. Stewart (1975); Lee and Fung (2008) calculate the equilibration as a function of the drop size) or to GCM parameterizations (e.g. Hoffmann et al. (1998) assume that only a proportion of the drops equilibrate in a saturated environment depending on the precipitation type).*

Thank you for asking to clarify this point. All drops equilibrate equally. In the case of saturated downdrafts, total equilibration is assumed, but below the cloud base the parameterization follows Stewart (1975), similar to other models mentioned by your comment.

p 8 l 30:

*How do you initialize water vapor composition at the model bottom and top? Interpolation needs these end members in addition to the tropopause values.*

These values are given in Table 1 and a reference to these values is included in the text.

p 9 l 9-14:

*Why is this paragraph here and not in the Methods section?*

It is not in the methods section because these are choices made at runtime and explicitly for these simulations. We have changed the introduction to the paragraph to make this more clear. We also spend more text explaining this in the introduction to the model (Sec. 2.1)

p 10 l 12:

*remove one "almost completely"*

Since we now use the decadal simulation and one year spin-up time, we have removed both "almost completely".

p 10, section 3.1:

*Previous studies using water tagging in models or water tracking tools should be cited and their results could be briefly compared to yours: e.g. Joussaume et al. (1984); Koster et al. (1986); Numaguti (1999); Yoshimura et al. (2004); van der Ent et al. (2010); Gimeno et al. (2012); Risi et al. (2013)*

Thank you for pointing out this literature. As suggested, we now cite four of these studies and compare the results to those of our paragraph.

p 18 l 13-18:

*but this does not improve the model-observations agreement…*

As measurements do not differentiate between vapor originating from land and ocean surfaces, our goal is not to improve the agreement between model results and measurements. The aim is to point out how the different tracers which are made available by ICON-ART-Iso can be combined for in-depth analysis of the results.

p 18:

*Can the lack of daily cycle be related to the wrong precipitation daily cycle, a known problem in many models (Betts and Jakob (2002); Guichard et al. (2004))?*

Thank you for pointing us at this idea. We have not followed up on it for the current manuscript, but may look at the daily cycle in precipitation in a future study.

p 20 l 9:

*How is this "sample" chosen?*

We now use the decadal simulation, which we sample along the paths of all those IAGOS-CARIBIC flights which took measurements of δD.

p 21 l 22:

*What explains the δD difference between these 2 parameterizations? Explain with simple physics what process is the main driver of this change. In addition, I understand that these 2 parameterizations differ for the representation of isotopic processes during rain evaporation, which occurs in the lower troposphere. Why does it have such a big impact in the upper troposphere?*

The difference between the two parameterizations is now discussed in App. B. Following comments by Referee 1, we revised part of the microphysics code in the model and redid all simulations. After reexamining the results with the revised model version, we decided to postpone the comparison of the two parameterizations to a later study in order to keep the paper concise.

p 22 fig 7.

*A few explanations on these distributions could be useful. For example, do you see the signature of condensate lofting/detrainement? In the upper tropospere, these processes are known to have a big impact on the isotopic composition (Moyer et al. (1996); Kuang et al. (2003); Bony et al. (2008); Sayres et al. (2010)). This could be discussed.*

A focused study is required to look into the details of these distributions. The investigation of ice lofting is one subject which could surely be addressed by use of these in situ data. It will be an interesting future study to investigate this with ICON-ART-Iso.

p 23 l 9:

*This is very vague: how can different processes be identified from a scatter? Be more precise.*

Agreed, it is probably impossible to read processes from one scatter plot alone. We have therefore removed the sentence and recombined the last two paragraphs of the section.

p 23 l 10:

*remove "of"*

corrected

Overall section 3.4.2 and previous sections:

*Be more quantitative when describing the model-observations agreement. Use quantitative metrics such as RMS error. This would allow to compare quantatively the model-observations agreement between different model version, different regions and seasons, different sampling criteria... It's difficult to assess the model-observations agreement by comparing by eye 2 different plots.*

Thank you for pointing this out clearly. We have included some numbers of mean values and offsets. In some cases, we also give relative differences.

p 23 l 5-8:

*Is the model-observations agreement better when removing these parcels with high proportion of initialization tracer?*

The main source for differences between model and measurements is the fact that the model is free-running. Removing the values with high proportion of initialization water therefore does not improve the agreement between model and measurements.

p 24 22:

*"discuss" -> show: you don't discuss the reasons for these differences.*

This paragraph has been changed, since we now also show these differences in the comparison of model results with the GNIP database.

p 24 l 32:

*"instances" -> use a more appropriate word?*

The word "instances" has been replaced by "tracers corresponding to".

**Reply to Referee 3**

Comments on

> From climatological to small scale applications: Simulating water isotopologues with ICON-ART-Iso (version 2.1)

Under revision at Geosci. Model Dev. Discuss., https://doi.org/10.5194/gmd-2017-280

Dear referee,

thank you for taking the time to review the manuscript. Your critical reading and the comments have made the text much more precise in many aspects. As proposed, we now include the dataset by LeGrande and Schmitt (2006) to prescribe the surface isotope ratio of the ocean.

Below, you will find a point by point reply to all your comments. We have attached a manuscript highlighting all the differences between the original and the updated version of the manuscript. This includes the changes from all three referees. For completeness, we have also attached the updated version of the manuscript.

On behalf of all coauthors,

Johannes Eckstein

**Major comments:**

Section 2.3:

> *There are two serious flaws in Section 2.3 that have to be corrected for to achieve a state-of-the-art water isotope enabled GCM:*

> *The authors are mistaken when they think that $R_{surf}$ is approximated by $R_{VSMOW}$ in Werner et al., 2011. In ECHAM5-wiso, the surface reservoirs soil, skin, snow and plant layer are filled by precipitation and form $R_{surf}$, which is then used for evaporation processes from the surface. Only fine processes like isotope fractionation during evapotranspiration are neglected. Using $R_{VSMOW}$ for the entire surface is a step backwards in global water isotope modelling.*

Thank you for clarifying this point about ECHAM5-wiso. The fact that we use this approximation of $R_{surf}$ is surely a big deficiency of the model. There are plans to implement water isotopologues into the surface module of ICON, but this is out of scope for this paper. In our analysis, we focus on comparisons using measurements taken over the ocean, which are less influenced by land surface evaporation. In addition, we often consider the evaporation tracers, which allow us to investigate the influence of land surface evaporation.

> *Also as a lower boundary condition for the ocean, this is not a state-of-the-art approximation and it is not done this way in recent models. Instead, a global gridded data set based on the 18O isotopic composition in sea water by LeGrande and Schmidt, 2006 (see attached figure) is taken e.g. in Werner et al., 2011. HDO in the ocean surface layer can be approximated from this.*

We have implemented the possibility to use this dataset for the isotope ratio of the ocean surface. Comments by referee 1 required us to change other parts of the code and to redo all simulations. For these simulations, we now use the proposed dataset to prescribe the ocean surface isotope ratios.

P6L5-6:

*Wrong. In fact, during freezing of water isotopic effects occur in a closed systems (Souchez and Jouzel, 1984) and there is even evidence of a kinetic effect (Souchez et al. 2000), but due to the comparably low diffusivities in liquid water these effects can be neglected in cloud processes in GCMs. This has to be mentioned somewhere, such inaccuracies should not make their way into scientific literature.*

Thank you for clarifying this point. We have included it in the manuscript and cite the corresponding literature.

Fig. 2 and Sect.3.2:

*You forgot to include a station in the tropics. Other processes (amount effect,...) become important here that have to be evaluated.*

Thank you for emphasizing this point. We have replaced Karlsruhe with Bangkok (13°N) to include a tropical station.

*The simulation setup and the initialization procedure is not suitable for the given goal of the study. To have a one-to-one comparison of water isotopologues in model results and observations a free running simulation with such short spin-up from such a crude initialisation is not applicable, but let me make these points one by one:*

*Since your aim is to evaluate your representation of water isotopologues (including the water cycle processes as well as fractionation effects) in the model and not the model meteorology, you first have to compare a nudged (specified dynamics) simulation. Otherwise you will never be able to separate the effects of an unequal meteorological situation with the water and isotope effects. And that is what you need to do in the first step, because only then you can really assess your water isotope implementation. Or the other way around, your evaluation is pointless with the used simulation.*

We are evaluating the water cycle of the non-hydrostatic model ICON. For this purpose, a free-running simulation is just as valuable as a nudged model run, while the focus may be slightly different in each case. In several previous studies, the validation of isotope implenetations has been done with a free-running simulation (e.g. Noone and Simmonds, 2002; Schmidt et al., 2005; Werner et al., 2011). In our free-running simulation, it is of value to see that the different seasonalities of the istopic composition of precipitation are well reproduced, just like they are in temperature.

On top of this, we investigate the statistical distributions of the isotopes. By showing how the model reproduces also these values, we are able to validate general aspects of the isotope implementation. Of course there are discrepancies that remain, but it will be the subject of future studies to investigate these and to improve the model.

Because of the explicit convection, non-hydrostatic models are generally more difficult to nudge than their hydrostatic predecessors. This is due to convective fluxes on the model grid, which are difficult to consider in the nudging process.

Taking these points into account, we are sure that the investigation with a free-running simulation is worth to be considered. In addition, we consider these results to be an important prerequisite for future studies with the model, the main reason for presenting the investigation in the manuscript.

*In particular with such a crude initialization of HDO, you need longer spin-up time, maybe a month or more. Then you will not face problems like in P21L31, so why not just do it? However, probably already the HDO field of your first simulation could serve as a (somewhat better) initialisation field for your second simulation.*

We now use a full year as spin-up time for the decadal simulation. The simulation ran from January 2007 to the end of 2017. By using the first of these eleven years as spin-up time, we still have a full decade to create the climatologies.

The section comparing CARIBIC measurements to the ICON-ART-Iso results uses a free running forecast of the model while attempting to reproduce the approximate meteorological situation at the time of two flights. Therefore, the simulation cannot be initialized with the meteorology of the decadal simulation. The influence of the initialization is discussed in this chapter by making use of the initialization water tracer.

Section 3.4:

*As mentioned above, the section on the comparison with CARIBIC data should be comprehensively restructured. Comparisons should be made point by point and clear conclusions should be drawn and presented out of that.*

In the updated manuscript, we have enriched this section by including several mean values and offsets of the different distributions to one another. We hope this makes the chapter more valuable.

*This brings me to my next point. The authors correctly write, that water isotope modelling is applied to answer climatological questions and (this should be "and" not "or" on page 2, see other REF) process understanding. This usually means the simulation of a particular phenomenon for in-depth analysis with the additional information from the water isotopes. Here, the authors are using the tool in a weather prediction setup, but they never clarify why they chose this configuration, i.e. how one could profit from this sort of "weather model setup". Or in other words, what advantages does this setup have over the application of the isotopes in a regular climate model configuration.*

ICON can be used for climate simulations as well as for weather prediction, the difference being only a matter of simulation time and output processing. For users of the model, there is no fundamental difference and the advantages you mention for the use of the isotopes in a climate model also apply for the weather model setup. Only that the capabilities of ICON are not limited to these application, but may well be applied to questions regarding shorter times scales and local phenomena, as we demonstrate in the later chapters of the manuscript. In addition, using the numerical weather prediction physics package provides us with the microphysical scheme by Seifert and Beheng (2006), which is well suited for case studies like the one we present in Sec. 3.4.

Section 3.2:

> *The concept of the tagged water tool is simple and it has been developed many times before (which the authors do not even mention), see REF#2.*

We now cite several of the studies suggested by referee 2 and compare the results to ours.

> *Thus, the concept itself does not require another reference in GMD. The process studies that are conducted with it in this paper do not provide any new insights into the hydrological cycle. Plus, there is only little connection of it to the actual topic of the paper, the only real connection is that one can approximate the spin-up time with the approach. This, however, the authors will not need anymore if they consider my point on the simulation setup (which is crucial for the evaluation and thus for the entire paper). Hence, this part of the study dangles somewhere in the nowhere here and should thus be removed completely from the paper.*

We now use the decadal R2B04 simulation for this investigation. We consider climatological summer (June, July, August) and winter months (December, January, February). The results therefore well complement the influence of ocean and land surface evaporation presented in the section comparing GNIP measurements to the model results. Thank you for motivating us on this point to more closely reconnect the section to the paper!

Even in a nudged simulation, a spin-up time is needed by the model to generate consistent isotope ratios. As described above, we believe this study to be an important prerequisite for future investigations, although the conclusions may be different than those drawn from a nudged simulation.

**Minor comments:**

P1L6:

> *This sentence needs to be rephrased, it is incorrect english and it is not clear how this shall represent a range of temporal scales.*

The sentence now reads: "The model is then evaluated on a range of temporal scales by comparing with measurements of precipitation and vapor.".

P2L7-9:

> *This sentence is grammatically wrong (dangling participle).*

The sentence now reads: "Considering the isotopologue ratio of the heavy isotopologues in vapor and precipitation (liquid or ice) provides an opportunity to develop an advanced understanding of the processes that shape the water cycle."

P2L26 and elsewhere:

> *Cauquoin and Risi, 2017 has been rejected by GMD. You may want to find another citation for this.*

Thank you for pointing this out. We have removed the citation.

P2L31:

*Most (if not all) models can be run with a fine horizontal resolution. How does the ICON model "stand out" from other models with respect to this?*

It is not the fact that the resolution can be increased which makes ICON and ICON-ART-Iso stand out, but the fact that it is a global, isotope-enabled model with a non-hydrostatic core. In addition, the numerical implementation makes it very efficient on modern computing systems. The ART modules – a fully integrated part of the ICON model – provide a flexible tracer infrastructure, which is used in ICON-ART-Iso to build a flexible isotope model. These are the characteristics that make ICON-ART-Iso stand out from other isotope-enabled circulation models and assure its long time use in the scientific community. We hope to have made this more clear in the updated version of the manuscript.

P2L32:

*You use the word "flexible" here and also later on in the paper, and I think the word is being misused here. A model is not flexible when you can implement a lot of diagnostics, or run several water tracers at once. Basically you can do that with every model. Flexibility would be for example when you can easily switch these diagnostics and tracers on and off or change their attributes without having to recompile. Or being able to expand the model system such that you have several co-existing processes (e.g. convection schemes) that can be run with the same executable. I am not aware if this is possible in ICON-ART-Iso, if so you could describe it and use the word flexible, if not, you should rather use the word comprehensive or extensive.*

Thank you for pointing out that this is not stated clearly enough in the manuscript. Since the isotopes are a diagnostic of the water cycle in the model, it is not possible to run several convection schemes in the same simulation. But your first point is exactly what we mean by flexible: Without recompiling, we can switch between different parameterizations, run several parameterizations for fractionation at the same time for different isotopes and even use different parameters for different isotopes that use the same parameterization. This concept is similar to what has been implemented for ICON-ART for chemical tracers and is explained by Schröter et al., 2018. We hope to have made this more clear in Sec. 2.1.

P2L32:

*You mention "diagnostic moisture tracers" and "tagged water" here, but at this stage the reader has no idea what you actually mean with that. Either explain it shortly, or wait until it comes to the point.*

The sentence now reads: "It is flexible in design to simulate diagnostic evaporation tracers as well as the isotopologues HDO and H218O during a single simulation."

P3L2 and L6 and throughout the manuscript:

*A section does not compare or discuss anything. These formulations make no sense, instead you compare it in the section. So use e.g.: In this section, we discuss….*

Thank you for making this point, we have changed the text accordingly.

P3L26:

*And why is this a very good assumption? Because the abundance of standard water is at least three orders of magnitude higher than the abundance of any rare isotopologue. Why not include this in the text?*

We now give the values.

P3L27-29:

*Rephrase: ...seven different forms (vapor, cloud water, ....), each of which is represented by one tracer for standard water and one tracer for the isotopologues HDO and H182O, respectively.*

We have taken up your suggestion and moved the list into brackets.

P4L1:

*What settings?*

We have expanded the paragraph to make more clear what we mean by settings, also see our answer to comment on P2L32.

P4L21-22:

*Sentence is unclear, rephrase.*

We have rephrased the sentence. It now states: "In order to turn one of the fractionating processes explained below into a non-fractionating process, its respective equation for the transfer rate of the heavy isotopologues can be replaced with Eq. 1. This has been implemented as an option in all processes that describe fractionation."

P8L23-24:

*At this stage, it is totally unclear where the simulation of "tagged water" is supposed to aim at, so this sentence is rather confusing.*

We have removed this part of the sentence.

P8L30 and throughout the manuscript:

*To avoid ambiguity, the δ should be complemented by the isotope, the molecule and if relevant also by the phase, such as: δD(H 2 O ice ).*

As is common when dealing with the heavy isotopes of water, all delta values are taken relative to H2O. We state this in Sec. 2.1 and now also specify that we always calculate it in the vapor phase (except in Sec. 3.2, where we make a statement about this).

Sect. 2.7:

*The text never says that you are using HDO measurments here (only in the table caption), that makes the read unnecessarily confusing*

The sentence now reads "Values for the tropopause level and the model top are taken from MIPAS measurements (Steinwagner et al., 2007), the value at the lowest level is a standard value taken from Gat (2010).", making clear that we use measurements.

P9L15-20:

*This paragraph leaves unclear what this is supposed to be good for, why do you do that?*

The initialization water allows us to see the influence of the initialization on the distribution of water vapor. Similarly, the ocean and land evaporation water tracers allow to investigate the influence the ocean and land surface each have on the distribution of vapor at a certain point of the simulation. We now state in the paragraph:
"Next to case studies, this is interesting because of the undercomplexity of the evaporation from land surfaces, see Sec. 2.3. The tracers of $q^{init}$ hold information of the importance of the initialization at a certain time in the simulation."

P9L22:

*Please rephrase, the sentence is unclear. Also, why are you doing that?*

As explained above, we now use the decadal R2B04 simulation also used to compare to GNIP measurements for this investigation. In addition, we do not use single months but we compare mean winter (December, January, February) and mean summer (June, July, August) values of ten years simulation.

Fig. 1:

*Give panels a,b,c,d,e,f and change text accordingly*

We have refrained from using panel numbers here as the panel title is clear enough.

P12L4:

*73h is a very coarse output time step. Why so coarse? That makes only 10 steps a month, I think that is too weak if you want a robust climatology.*

In the simulation now used for the investigation, we have reduced the output time step to 10 hours.

P12L5:

*Why 91.25? Magic number?*

This number came from not considering the first 30 output fields, which were 73 hours apart. This makes a total of 2190 hours or 91.25 days of spin-up time. In the evaluation of the new simulation, we use a full year as spin-up.

P12L12-13:

*Shortly explain why you do that.*

The additional evaporation tracers allow a deeper insight into the water cycle by showing the importance of land and ocean surface evaporation for each station in each month.

P12L27-32:

*The temperature bias should have a large impact on the isotope ratios. That does not always seem to be the case, see .e.g. Ankara. How come?*

There is surely a relationship between the surface (2m) temperature and the isotope ratios in precipitation. But there are also many other factors that strongly influence this ratio. The isotope ratio in precipitation is the result of all processes that lead to the formation of this precipitating water. To disentangle what influence the surface temperature has on these ratios is beyond the scope of this paper, but it will surely be considered in future studies.

*In the depiction it is hardly possible to see how far the values are off, the vertical axes are too coarse.*

Thank you for pointing this out. We have changed the vertical axes, synchronizing now for all stations but Halley Bay. This makes it easier to see differences between the stations and between the model and measurements.

P14L25:

*Why 90%? Another magic number?*

The cloud variable used for the threshold quickly reaches 100%. It therefore is reasonable to use a threshold of 90% to exclude all cloudy points. We have also changed the text, explaining this matter.

Sect. 3.3.1:

*Do you take the model output at the same times and the same locations as the satellite makes observations (overpasses)? For this comparison you should.*

We do not compare model data collocated to the measurements, but this is also not necessary. In a free-running simulation, we do not expect to get the same meteorological situation that the satellite has seen. Taking only collocated points is therefore unnecessary. We do, however, use the same regions that are used for the evaluation of the satellite data.

Fig. 3 and 4:

*Give panels a,b,c,d and change text accordingly*

Again, we have refrained from numbering panels when the title of each panel is very clear.

P18L4:

*Two more magic numbers, please explain.*

These numbers were chosen to extract datapoints that have a strong characteristic of ocean or land surface evaporation. Therefore, it is difficult to give globally valid numbers here or even numbers that are reasonable for the ocean or land in general. If ocean evaporation water reaches 90% over the ocean, the sample can be considered characteristic. (99% would be even more characteristic, of course) But over the dry Saharan desert, it is close to impossible to find water vapor made up of 90% land evaporation water. Therefore 50% is a good number here. We then use the same numbers to find characteristics in the tropical samples.

Of course these numbers are arbitrary to a certain degree. But here, they serve to illustrate a method rather than being the basis for a scientific conclusion. The combination of water evaporation tracers with isotope values shows how the two can go hand in hand to investigate moisture pathways. In the previous section, all water vapor is considered, while we here consider only datapoints characteristic of a certain source region. As is shown in the figures, the distributions differ strongly. We believe this can be developed into a powerful tool in future studies, which is the reason for presenting the method in the manuscript.

P19L16-17:

*That would mean the flights take place in the extratropics only, because in the tropics these altitudes are still the troposphere. The flights do however seem to cross the equator, something is wrong here!*

The flights do not cross the equator, see Fig. 6 of the GMDD manuscript. The method for deriving δD described in the manuscript is also used for the tropics.

P19L26:

*What conditions? Weather in general, or humidity, or what?*

The two flights – only hours apart – sampled air of a similar meteorology: the outflow region of tropical storm. The same is true for the model sample along the flight tracks. The model does not reproduce the storm exactly – as we use a forecast run – but the general situation is similar.

As in the previous sections, we do not attempt to disentangle all processes happening here, but rather show the general performance of the model in reproducing the upper tropospheric distribution of δD in vapor. In addition, we discuss how small scale processes in the model are imprinted on the isotope signal. But here, we do not compare measurements directly, well aware that this is not possible with a single, free-running forecast simulation.

P20L4-5:

*The flights do not sample the model atmosphere, the flights take measurements of the real atmosphere. Your off-line interpolation of the model data samples the model atmosphere. Be more precise.*

As suggested, we have reformulated the paragraph, also removing this inaccuracy.

*Moreover, taking only 1h output for this comparison you should make clear that, with an approximate flight velocity of the aircraft of around 900km/h, you will only get less than 10 model points per flight you are interpolating your data to. That is very coarse given that you are aiming on comparing local phenomena. In fact, it would be better to use an on-line flight tracking tool with high temporal resolution for this.*

An on-line interpolation tool is currently not available for ICON. For the simulation we use now, we have reduced the output frequency to 1 per 15 minutes and use an output resolution of 0.5° or roughly 55km on the equator and less further north. This is close to the numeric grid resolution used here (R2B06), which corresponds to about 40km resolution globally. In 15 minutes, there are about 15 measurements along roughly 225km (distance of

4 grid points). In a linear interpolation, at least 8 spatial grid points provide information. Since we also do a linear interpolation in time, there are at least 16 model values which provide information for the 15 measurements. We believe this captures the relevant features of the model, especially considering that in the end, we compare an aggregated distribution to the CARIBIC measurements.

Sect 3.4.1:

*Please state when exactly was which flight.*

CARIBIC flight 309 from Frankfurt to Caracas took place from September 22, 2010 10:16 UTC to September 22, 2010, 19:32 UTC. Flight 310 back from Caracas to Frankfurt started on September 22, 2010, 22:12 UTC and landed on September 23, 2010 07:31 UTC. We now give the exact departure times in the text.

P20L7:

*The hurricane processed your data? Please rephrase.*

We now state "In order to compare values influenced by hurricane Igor..."

P20L8:

*What model simulation?*

As explained in the updated manuscript, we now use the decadal R2B04 simulation for the tropical sample. We interpolate the output of this simulation to all CARIBIC flight paths and then reproduce the distribution of tropical data from these values.

P20L9:

*What exactly do you mean by "the IAGOS-CARABIC database is examined"?*

We have changed the wording of the sentence: "For reference, all tropical δD values from the IAGOS-CARIBIC database are also examined."

P21L1:

*I do not understand what you mean by "randomly drawn" and "randomly chosen", please explain. If you want to make a climatological comparison here, why not make a long simulation to have a fair comparison?*

As explained above, we now use the decadal R2B04 simulation. We interpolate the output of this simulation to all CARIBIC flight paths and then reproduce the distribution of tropical data from these values.

P21L23-24:

*From this depiction I cannot even see this. You should make the evaluation much more quantitative.*

As also suggested by referee 2, we have included mean values, absolute and relative offsets where necessary in the revised version of this chapter.

P21L31:

*This is one part of what I mean with my major point on the simulation setup. If you had a longer (some months) spin-up time, you would not have to face these problems.*

A longer spin-up time would make it impossible to compare to the measurements taken during the event of Hurricane Igor since the model is free-running. We consider the influence of the initialization by means of the initialization water tracer. By the time of the flights, about 50% of the water sampled in the model is influenced by the initialization.

Fig. 7:

*What does the N stand for?*

The N indicates the number of data points used for the creation of the distributions. We now indicate this in the figure caption.

P23L5-8:

*So if you want to discuss stratospheric intrusions you can not get around methane oxidation and its influence on δD(H2O) in the stratosphere. Depending on from which altitude the air is transported downwards, this could mean strongly depleted or enriched water vapour. Also, work has already been done to parameterize this effect on HDO in GCMs (Schmidt et al. 2005, Eichinger et al. 2015), it is a shame that this is not even discussed here given that you are evaluating results in the UTLS.*

Thank you for pointing this out. Since the flights no longer see a stratospheric intrusion in the new simulation (there have also been updates to other parts of the model, which change the dynamics), it has become unnecessary to discuss these here.

P23L8:

*"air has been processed by the model" What does that mean? Please rephrase.*

We now state: "...indicating that the model atmosphere has seen many fractionating and transport processes. This includes air mass mixing, but also the microphysical and convective processes that are imprinted on the isotopologue ratio. The exact nature of these processes remain to be investigated by future studies."

P23L9:

*"different processes" What processes? At least shortly list some.*

See reply to the previous comment on P23L8.

Sect. 4:

*Change title from "Conclusions" to "Summary (and outlook)", there are no conclusions that go further than what had already been written before.*

As this is a technical article, it is the nature of conclusions to summarize more than to interpret results. We have therefore decided to leave the section title to read "Conclusions".

P24L10:

*What do you mean by "long term stability"?*

In order to make the sentence more concise, we have removed this part of it and it now reads: "This investigation presents a first climatological application."

**Technical issues:**

| | |
|---|---|
| P1L18: | *...oceans are an unmatched …* |
| | Corrected to "...the oceans are unmatched water reservoirs, which dissolve trace substances (Jacob, 1999) and redistribute heat (Pinet, 1993)..." |
| P1L18: | *"a reservoir to dissolve" - refine wording* |
| | see above reply to comment on P1L18 |
| P1L12-13: | *It is not clear what is meant by "all of tropical data" here.* |
| | Corrected to: "The general features of this sample as well as those of all tropical data available from IAGOS-CARIBIC are captured by the model." |
| P2L8: | *... isotopologues in water vapor ... - this inaccuracy appears several times* |
| | We believe that the context makes clear enough that *vapor* is is *water vapor* here. |
| P2L14-15: | *Refine the english. Suggestion: The isotopologue content of water vapor has first been measured by means of cryogenic samplers (Dansgaard, 1964). In the last 15 years, also laser absoption spectroscopy methods have been developed for that use.* |
| | We have followed the suggestion of referee 1. The sentence now reads: "Measurements of the isotopic content of vapor first required cryogenic samplers (Dansgaard, 1954), but in the last 15 years laser absorption spectroscopy has made in situ observations possible (Lee et al., 2005; Dyroff et al., 2010)." |
| P3L7: | *... with the results of ICON-ART-Iso simulations. (You compare the results, not the simulations)* |
| | corrected |
| P3L14: | *... is the water isotopologue enabled …* |
| | We have decided to write about *isotopologues* and not *isotopes*. Still, we keep with the generally accepted and shorter term of an *isotope-enabled model*. |
| P3L25: | *To discriminate .... , unclear, rephrase.* |
| | The index $h$ is used to make variables of the heavy isotopologues different to those of standard water. |
| P3L26: | *For clarity, you should write 1H162O here.* |
| | Following your suggestion, we spell out the exact isotopic composition here once. |
| P4L1: | *Remove "As".* |
| | We have replace "As in..." with "Similar to..." |
| P4L17 | *The parameterization that influence …* |
| | We have corrected this by including "which". |

P4L29-30:      *The ratio of ratios?*

Yes, α can be seen as a ratio of ratios.

P4L31 (and throughout the section): *Put equation 2 here, not at the end of the paragraph.*
*Generally restructure the presentation of equations.*

Following your comment and several comments by referee 1, we have restructured all sections explaining the physics of fractionation.

P6L6:      *ilead*

corrected

P8L20:      *... in the standard setup*

corrected

P8L23:      *initialization of the water isotopologues!*

The first sentence of Sec. 2.7 now reads: "A meaningful initialization is an important prerequisite for any simulation, also of the isotopologues."

P9L9-14:      *repetition*

This paragraph is not a repetition but gives the settings for the simulations which are presented in the following sections. We hope this is more clear with the restructured Sec. 2.1.

P9L15:      *what are water species?*

We have changed "diagnostic water species" to "diagnostic sets of water tracers".

P9L21:      *diagnostic*

corrected

P13L3:      *remove "In doing so"*

removed as suggested

P18L15:      *criterion*

corrected

P19L8:      *... in the tropics serves as reference.*

Corrected and now reads: "...taken by IAGOS-CARIBIC in the tropics is also used as reference."

P19L23:      *...resolution is finer than the …*

Following your suggestion, we have replaced "exceeds" with "is finer than".

P19L24:      *remove "very"*

corrected

P19L28:      *... Hurricane Igor had passed …*

As suggested, we have included "had".

P21L5:      *laong*

corrected

P21L6:      *deld*

corrected

P23L5:      *show show*

The sentence has been removed.

P23L10: *...along the …*

The sentence has been removed.

P23L16: *remove "of"*

corrected

[revised manuscript text omitted]